# A Geospatial Decision Support System for Supporting the Assessment of Land Degradation in Europe

Piero Manna [1,*], Antonietta Agrillo [1], Marialaura Bancheri [1], Marco Di Leginio [2], Giuliano Ferraro [3], Giuliano Langella [3,4], Florindo Antonio Mileti [3], Nicola Riitano [2] and Michele Munafò [2]

[1] Institute for Mediterranean Agricultural and Forestry Systems (ISAFOM), National Research Council (CNR), 80055 Portici, Italy; antonietta.agrillo@unina.it (A.A.); marialaura.bancheri@cnr.it (M.B.)
[2] Italian Institute for Environmental Protection and Research (ISPRA), 00144 Rome, Italy; marco.dileginio@isprambiente.it (M.D.L.); nicola.riitano@isprambiente.it (N.R.); michele.munafo@isprambiente.it (M.M.)
[3] CRISP Research Center, Department of Agriculture, University of Napoli Federico II, 80055 Naples, Italy; ferraro.giuliano@gmail.com (G.F.); glangella@unina.it (G.L.); florindoantonio.mileti@unina.it (F.A.M.)
[4] Department of Agriculture, University of Napoli Federico II, 80055 Naples, Italy
*  Correspondence: piero.manna@cnr.it

**Abstract:** Nowadays, Land Degradation Neutrality (LDN) is on the political agenda as one of the main objectives in order to respond to the increasing degradation processes affecting soils and territories. Nevertheless, proper implementation of environmental policies is very difficult due to a lack of the operational, reliable and easily usable tools necessary to support political decisions when identifying problems, defining the causes of degradation and helping to find possible solutions. It is within this framework that this paper attempts to demonstrate a new valuable web-based operational LDN tool as a component of an already running Spatial Decision Support System (S-DSS) developed on a Geospatial Cyberinfrastructure (GCI). The tool could be offered to EU administrative units (e.g., municipalities) so that they may better evaluate the state and the impact of land degradation in their territories. The S-DSS supports the acquisition, management and processing of both static and dynamic data, together with data visualization and on-the-fly computing, in order to perform modelling, all of which is potentially accessible via the Web. The land degradation data utilized to develop the LDN tool refer to the SDG 15.3.1 indicator and were obtained from a platform named Trends.Earth, designed to monitor land change by using earth observations, and post-processed to correct some of the major artefacts relating to urban areas. The tool is designed to support land planning and management by producing data, statistics, reports and maps for any EU area of interest. The tool will be demonstrated through a short selection of practical case studies, where data, tables and stats are provided to challenge land degradation at different spatial extents. Currently, there are WEBGIS systems to visualize land degradation maps but—to our knowledge—this is the first S-DSS tool enabling customized LDN reporting at any NUTS (nomenclature of territorial units for statistics) level for the entire EU territory.

**Keywords:** land degradation; LDN; NDVI; environmental awareness; spatial decision support system; modelling





## 1. Introduction

Soil provides presently irreplaceable ecosystem services such as food production, water filtration and climate regulation [1,2] and it is continuously threatened by land degradation processes which reduce its capacity to support human and other life on Earth [3].

In recent decades, soil degradation processes have increased significantly. Processes such as erosion, organic matter decline, compaction, salinization, sealing, landslides and contamination are likely to accelerate soil degradation further if nothing is done to protect this resource [4]. A recent review by the EU Soil Health and Food Mission Board and

JRC [5] showed that approximately 60–70% of soils were in an unhealthy state in 2020 (what is more, this figure underestimates soil pollution).

On the policy side, a solid push to fight such soil and land degradation comes from the well-known United Nations 2030 agenda, where 7 of the 17 Sustainable Development Goals (SDG) refer to soil as a resource to be protected and maintained [6]. More specifically, the SDG 15.3 target has been adopted to combat desertification, restore degraded land and soil, and achieve a land degradation-neutral world. To achieve this goal, the SDG 15.3.1 indicator has been established (UNCCD, 1994) to monitor the "proportion of land that is degraded over total land area" and as a basis to plan actions and investments to reverse land degradation. In addition, the UNCCD rationale highlights that in performing SDG 15.3.1 estimates, it is essential to use, where possible, local dataset and the most appropriate methodology as described in the SDG 15.3.1 Good Practice Guidance [7].

To enable uniform SDG 15.3.1 indicator assessment, the UNCCD, based on previous work (Inter-Agency and Expert Group on Sustainable Development Indicators in November 2017), defines the following three sub-indicators: vegetation productivity, land cover changes, and soil organic carbon changes. These indicators aim to quantify the ability of the land to provide ecosystem services [8]. Typically, they are calculated and combined by using the "one-out-all-out" methodology (1OAO) proposed by the Trends.Earth software [9], which processes raster dataset (e.g., MODIS, AVHRR, CLC) and allows a pixel-based qualitative index of potential degradation (see materials and methods).

Indeed, the proposed indicators are useful tools because they allow us to have an explicit spatial picture of soil and land conditions over time. Nevertheless, there is increasing evidence that the SDG 15.3.1 indicator approach has some embedded reliability problems, possibly causing dispute and misleading policy guidance i.e., [10].

Among these problems are such aspects as the fact that: (i) the three indicators have many data products [11], and thus Land Degradation values calculated by using these different products may not be consistent [12], so posing an important uncertainty problem. (ii) There is also a well-known issue about using either one (e.g., NDVI) or a number of vegetation indexes (e.g., NDVI, EVI, LAI, NPP) [13,14] in order to handle both vegetation productivity dynamics and phenology variability properly [15]; thus, the current Trends.Earth oversimplifications in using vegetation indices can lead to misunderstanding and, above all, there is still no consensus on selecting appropriate indicators [14]. (iii) The impact of drought variability on vegetation indices poses a great problem in assessing land degradation [16]. (iv) The need for a finer-resolution dataset is generally essential, but even more so in order to set accurate baseline indicators in the initial SDG 15.3.1 tracking year [10]. (v) The lack of a well-established quantitative relationship between the SOC stock (soil organic carbon stock), delivery of ecosystem services and connected land and soil degradation [17]. (vi) The oversimplistic classification (strongly empirical) into degraded and non-degraded land [11] while [18], for instance, have demonstrated that the combination of LDN assessment with wildfire, landslides, and drought areas is performed much better by identifying 12 land degradation categories. (vii) The use of LDN administrative boundaries for assessing and reporting degradation suffers from imprecise estimates especially when addressing a mosaic of biomes [10]. (viii) Procedures to detect and exclude non-anthropogenic processes are absent from the SDG15.3.1 recommendations for measurement [19].

The occurrence of many detailed studies which report limitations in LDN estimates demonstrates the increasing importance of land degradation in science. More specifically, over the last two decades, there has been an exponential increase in scientific publications on land degradation assessment methodologies and new approaches [20]. Thus, the entire research area is very dynamic and new SDG15.3 approaches must be able to adopt any new improvement quickly once it has been properly validated.

Above all, long-term remote sensing data on vegetation productivity over time is still the only suitable approach to monitoring productivity degradation from local to global scales.

Therefore, current limitations have to be dealt with while we slowly make progress with the science to ameliorate SDG 15.3.1 assessment and better communicate [21] the critical importance of LDN at any administrative level and to the general public.

In general terms, it can be stated that there is a need to improve SDG 15.3.1 assessment approaches so enabling us to achieve both feasible, easy, sustainable uploads of new data/model produced by better scientific approaches or simply by a better local dataset (UNCCD) and easy communication towards large public and public bodies through easy SDG 15.3.1 data access, visualization and reporting at local, regional, and national levels.

The good news is that the research community is moving in these directions, as evidenced by numerous and recent scientific papers devoted to the issue, such as [22], by which the authors propose a proof of concept for a scalable and flexible approach to monitor land degradation at various scales (e.g., national, regional, global) using various components of the Global Earth Observation System of Systems (GEOSS), or [23] in which the authors address the reliability of UNCCD indicators results when compared with perception-based studies, and the implications this has for setting and monitoring land degradation.

Interesting FAO projects also address the issues of scalability and resolution through the creation of web-GIS systems in which the main goal is to allow easy visualizations of global SDG products based on different algorithms, raising awareness about the need to create products at the national level that fit local conditions (https://projectgeffao.users.earthengine.app/view/reu-ldn-assessment "URL accessed on 15 December 2023"; https://wocatapps.users.earthengine.app/view/ldn-prais4 "URL accessed on 15 December 2023").

Other examples of such web-GIS systems are *LandGis* (https://opengeohub.org/article/pre-release-landgis/ (accessed on 15 December 2023)) presented as the "OpenStreetMap for land-related environmental data", *Queensland soil and land resource data web map service* (https://www.data.qld.gov.au/dataset/queensland-soil-and-land-resource-data-web-map-service (accessed on 15 December 2023)) which contains soil site data and soil polygon mapping data which is represented as four main types: soil, land management and land degradation mapping, *Land Portal* (https://landportal.org/book/indicator/un-aglnddgrd (accessed on 15 December 2023)) expressly dedicated to several land indicators including the SDG 15.3.1, However, in this entire domain, there is a clear knowledge gap between the potential of the SDG15.3.1 indicators and their actual use in land planning and management. In addition, the potential use of these indicators is limited if they are not integrated into operational support systems which produce spatially explicit information and provide more structured indications (e.g., location and causes of and use of the land affected by the potential degradation) to better implement environmental policies.

Based on this background, the work proposed has a double objective: (i) a reliability assessment of the Trends.Earth approach limited to urbanized areas, followed by a potential improvement in the SDG 15.3.1 indicator and (ii) the development of an operational LDN tool to enhance SDG 15.3.1 applicability towards an extensive range of potential users. In fact currently there are WEBGIS aiming to visualize land degradation maps, as described above, but there is not a SDSS tool enabling customized LDN reporting at any NUTS level for the entire EU territory.

The LDN tool has been developed on a new type of already running Spatial Decision Support System, developed within an Horizon2020 project named LANDSUPPORT (www.landsupport.eu; "URL accessed on 10 June 2023") [24]. The tool aims to provide a web-based operational tool to challenge land degradation at different EU territorial units, from country to municipality level (NUTS 2,3,4), by applying the SDG15.3.1 indicator.

The tool will also be described by reporting some case study applications.

*The LANDSUPPORT Platform*

The LDN tool is part of the LANDSUPPORT platform, along with other tools for different features at different scales and domains relating to forestry, agriculture and other environmental issues [25–27]. LANDSUPPORT users can run the tools via browser by creating an account for free at https://app.landsupport.eu/ ("URL accessed on 10 June

2023"). They can visualize and download maps, charts and data and submit elaborations with different levels of complexity. Some pre-processed data are already provided on the platform, whereas much other processing is executed on the fly. The platform is developed within a Geospatial Cyber-Infrastructure (GCI) [27], which can acquire and process both static (e.g., administrative boundaries) and dynamic data (e.g., data varying over time such as population, land cover or weather data).

Thanks to a 3-tier logical architecture described below, users can interact with digital maps, charts and table data through server-implemented applications, sometimes made up of complex rules (Figure 1).

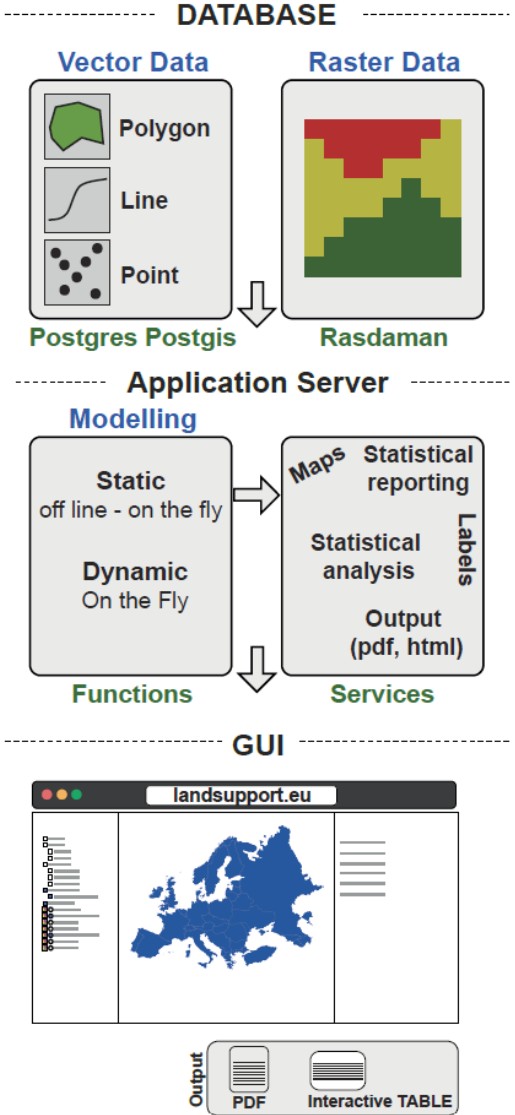

**Figure 1.** Synthetic workflow of the basic structure of the LANDSUPPORT GCI architecture, functions and technological components.

The geo-database includes vector and raster data. Vector data are stored using an open-source license database (PostgreSQL 9.5.16 and PostGIS 2.5), which allows the management of both spatial and non-spatial data, while the Rasdaman datacube technology [28] handles, manages, stores and recovers huge multidimensional arrays of raster data. The server side includes services, processes, flows and functionalities to ensure the overall execution of the system and calls the appropriate modelling chain according to users requests. Indeed, it is the component of the GCI where the requests submitted by the end-users are dealt with. GUI (Graphical User Interface) includes graphical tools, procedures for combining

spatial data (analysis and visualization), the production of tables and maps, and easy and intuitive navigation.

The interface has three sectors: "data viewer", "map viewer" and "analysis tools", which are described in the illustration of the LDN tool below.

The activation of each tool will run several processing procedures within a specific region of interest (ROI), ranging from simple visualizations of thematic maps with the option of running geospatial operations to more complex elaborations involving the applying of models on different spatial scales.

According to the tasks requested, three types of model can be executed: (i) static models using off-line processing and uploading onto the server output which is ready to be used in series by another model (codes) or visualized (e.g., maps and tables); (ii) dynamic models which give the users the opportunity to draw on-line their ROI, within which modelling procedures guarantee, on-the-fly, operations such as zonal statistics; (iii) statistical models operating calculations such as mean, max, min, standard and deviation, as well as spatial processing routines to calculate parameters within specific ROI (e.g., main land use/land cover classes).

As described in detail in the results section, the LDN tool has been developed by exploiting models i and iii.

## 2. Materials and Methods

*The SDG 15.3.1 Indicator*

The dataset linked with the LDN tool includes geo-referenced data and metadata, in both raster and vector formats, from different sources. Some of these data were already available and are included here as support layers. Some required further processing to be loaded into the DSS, while many others have had to be created as new data (see results section). ETRS89 was chosen as the European reference system (EPSG:3035), given the need to measure surfaces on different NUTS levels.

Before being integrated into the database, all spatial data were checked for potential anomalies. Vector layers (polygons) were firstly checked and eventually corrected in GIS environment for geo-referencing and data anomalies (i.e., spatial coordinates, missing data, outliers) and then loaded into the geospatial database, which allows location queries (run in SQL).

The most important layer for the LDN tool is the land degradation map, which was chosen to assess the state of land for the entire European territory through the SDG 15.3.1 indicator. The methodology adopted is based on the UNCCD recommendations in the update to GPG SDG indicator 3.5.1 [29]. This methodology involves the calculation/use of three indicators (land productivity, land cover changes and soil organic carbon changes) by applying Trends.Earth software 2.1.8 (http://trends.earth/docs/en/ "URL accessed on 10 May 2023"). Evaluation of the land productivity indicator is based on the calculation of annual integrals of NDVI (at 250 m cell size) by using bi-weekly products from MODIS and AVHRR satellite imagery; The indicator is then assessed by using three measures of NDVI time series change (sub-indicators): Trajectory, State and Performance. Time series are potentially variable according to the objectives and data availability. In this context, the degradation map integrated into the database was calculated by considering the time window 2001–2018.

Trajectory measures the rate of change in productivity over time. Trends.Earth computes a pixel based linear regression of mean values of annual NDVI to identify areas experiencing changes in productivity. Here, according to statistical tests (Mann–Kendall), positive trends in NDVI would indicate potential improvement, whereas consistent vegetation decline is interpreted as a signal of land degradation [30].

The State sub-indicator allows detection of recent changes (2015–2018) in primary productivity compared with a baseline period (2001–2014).

Finally, the Performance sub-indicator compares local productivity (NDVI mean values for a specific time period) with the productivity of other similar combinations of soil

units (USDA System provided by SoilGrids at 250 m resolution) [31] and land cover (ESA CCI layer at 300 m resolution).

The three sub-indicators are then combined following a "one out all out logic" (1OAO) to calculate the land productivity indicator.

The land cover changes indicator requires a homogeneous coverage over the study area to evaluate baseline and target periods. To this end were used Corine land cover maps for 2000 and 2018 (CLC), in raster format with 100 m cell size, using as common land cover legend/classes for the indicator the CLC first level classes of forest and semi-natural areas, agricultural areas, wetland, artificial surfaces and water bodies. Land cover transitions to be associated with degradation or improvement are classified according to a transition matrix [9].

The soil organic carbon changes indicator quantifies changes in soil organic carbon (SOC) over the same reporting period (2000 and 2018). A combined SOC/land cover method is used to simulate SOC changes in the first 30 cm. SoilGrids (250 m cell size) [31] is used as a reference estimate for the content in tons per hectare of organic carbon, while CLC time series (2000–2018) are used for land use change detection.

Changes in SOC due to land cover transitions are calculated with different coefficients for each of the main global climatic regions, as suggested by the methodology proposed by Conservation International. Negative changes in the carbon content equal to or higher than 10% are considered as potential degradation while gains in SOC content equal to or higher than 10% are considered as potential improvement in soil conditions.

Finally, the raster map of the land degradation indicator (SDG 15.3.1) is calculated according to a (1OAO) logic, i.e., assigning the state of degradation to all those areas in which at least one of the three conditions presented above worsened significantly during the reporting period. Following these rules, all other areas are classified as potentially improved or stable (Figures 2 and 3).

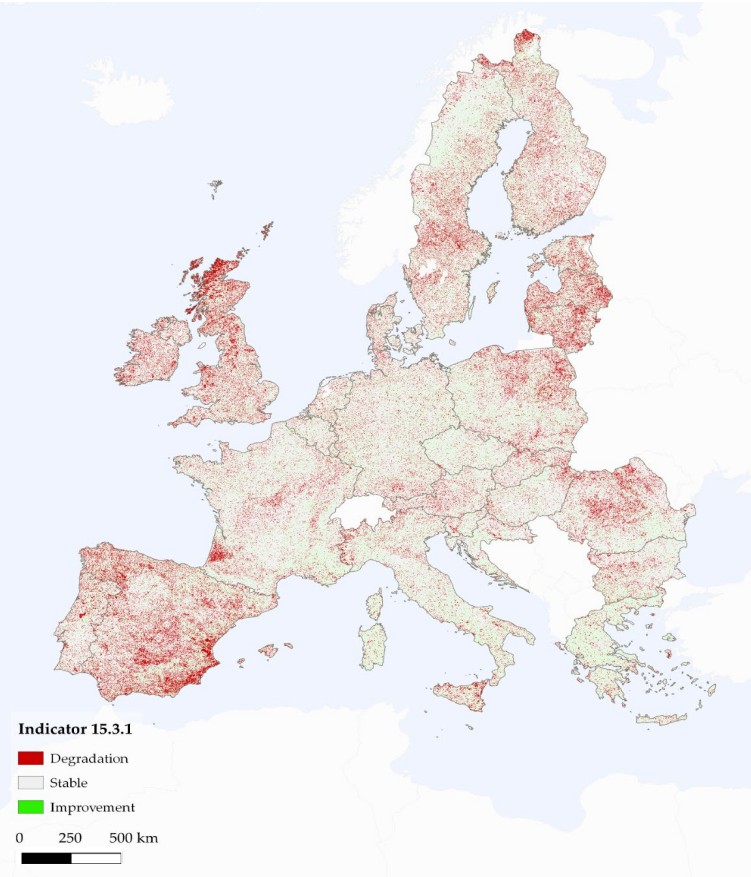

**Figure 2.** SDG 15.3.1 indicator. Assessment for Europe (reference period 2001–2018). Red pixels: areas classified as degraded; green pixels: areas classified as improved; no color: areas classified as stable.

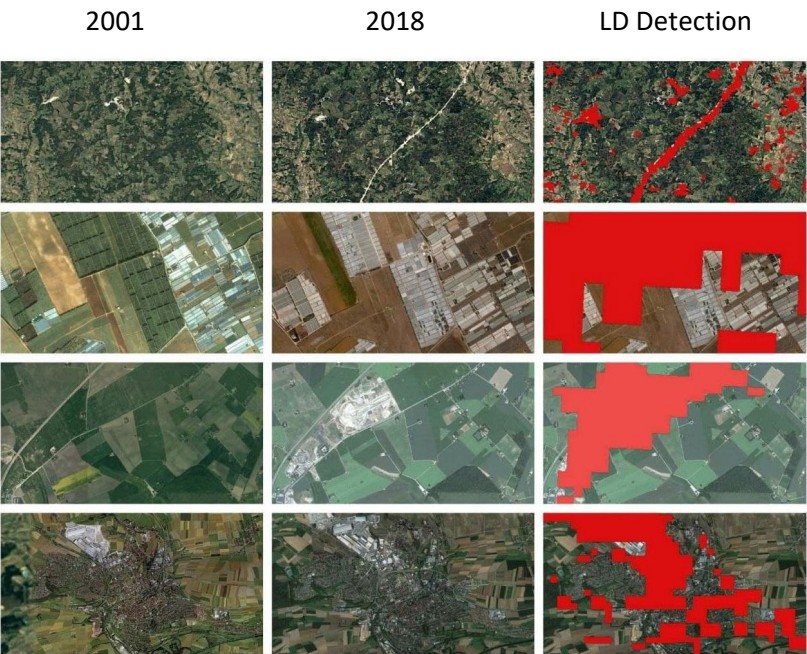

**Figure 3.** SDG 15.3.1 indicator. Details of change detection between 2001 and 2018. Red pixels: areas classified as degraded. No color: areas classified as stable.

It is important to stress here that, once produced, the SDG 15.3.1 raster indicator was first checked for anomalies (i.e., geo-referencing, projection), then its reliability in urban environments was improved and, finally, it was loaded into the LANDSUPPORT geo-database [24] to be used as basic data for the LDN tool.

## 3. Results

### 3.1. Indicator Test, Verification and Improvement

The SDG 15.3.1 indicator was checked for reliability by observing the results it produced in areas where there was certainty that the surfaces remained stable over time. Some of such easy-to-be-treated cases are the historical centers in urbanized regions (since already urban areas could not undergo any further land degradation).Google Earth satellite orthophotos observed with the time scrolling tool were used as further verification that there were no changes in the areas chosen during the period covered by the indicator. Then, since Trends.Earth does not allow these trends to be observed, the SDG indicator was reproduced by applying Google Earth Engine (GEE) codes so as to analyze NDVI time series in greater detail. The trajectory codes were wrote and the graphs of NDVI trends at specific locations for the period 2001–2018 were produced. Then, the codes were launched after selecting some historical centers of urbanized areas (Figure 4).

This approach highlighted a critical issue and errors relating to the classification procedures in the form of misclassification in many urban environments where stable surfaces were classified as improved or degraded. This is shown in Figure 4 where, for urban areas that were not subjected to new sealing or de-sealing processes (at least for the period and resolution considered by the indicator), very slight variations in the slope of the trend are classified as phases of improvement or degradation. Moreover, these variations occurs in an NDVI range that is very narrow, slightly higher or lower than 0.2, and such artefacts were found to be widely spread. Indeed the indicator classified as degraded or improved extended surfaces within the historic centers of big cities all over the EU, such as Berlin or Rome. Most probably these artefacts can be attributed to excessive sensitivity of the NDVI trend classification method and to the imprecise spatial resolution of the satellite data, which is potentially influenced by small scattered green urban areas. These evidences are somehow confirmed in a work published on an evaluation of the

indicator 15.3.1 performance [20] in which the authors propose conduction of case studies with high-resolution data and setting of local thresholds, but further studies are required, especially with regard to all the other types of land cover.

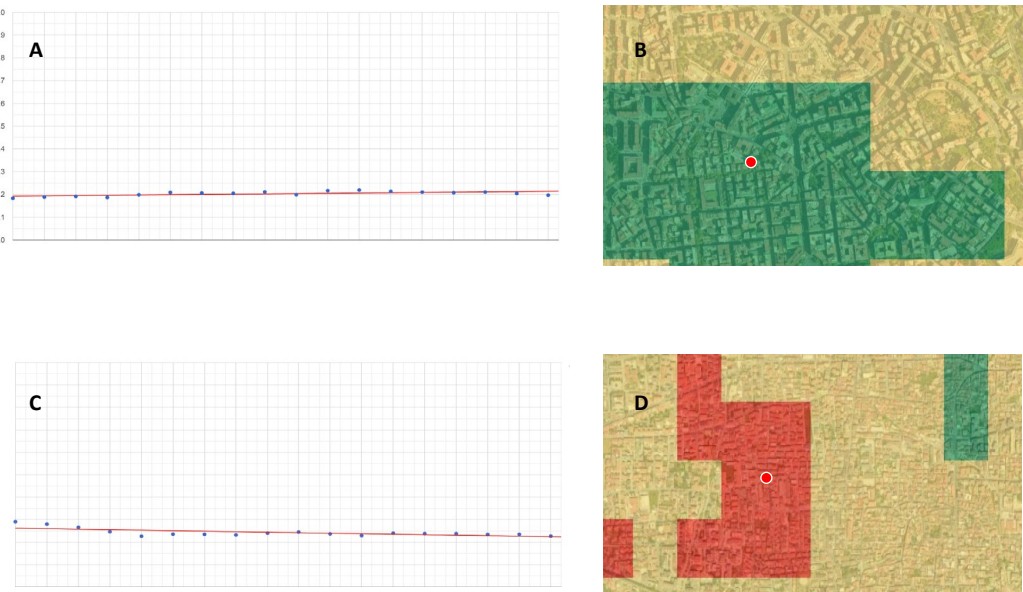

**Figure 4.** Two examples of NDVI trend analysis superimposed over urbanized areas (visible behind) by SDG indicator classes. Legend: green (improvement), yellow (stable), red (degraded). Red dots: centroids of pixels where the GEE codes have been run. (**A**): positive trend as classified in (**B**); (**C**): negative trend as classified in (**D**).

### 3.2. The Improved SDG 15.3.1 Indicator

On the basis of the above findings, the data for urban areas were improved by applying a procedure in a GIS environment. The procedure consisted of assigning the "stable" class (pixel value = 0) to all urban areas that were not subject to changes in the time range covered by the indicator (2001–2018). These areas were identified by matching the raster maps of the CLC 2000–2018 for the urbanized class (CLC level 1) so as to extract the pixels that were stable. The pixels extracted represented a mask to be applied to the indicator. Even considering that this procedure covered compact urban areas and historic city centers, which at the spatial scale of the indicator are actually almost totally sealed surfaces, the authors acknowledge that CLC1s for urban areas—used as a measure of soil sealing—typically produce an overestimation (e.g., green infrastructure within the urban center is considered urban) [32]. But this exaggeration was considered an advantage for this specific work because it ensured no misclassification in urban areas. Indeed, the application of the mask significantly improved the reliability of the indicator in EU urban environments as shown below.

Figure 5 shows the municipality of Rome in Italy (white boundaries) where red surfaces represent the surfaces falling within the urban class (CLC level 1) considered "stable" from 2000 to 2018; Red surfaces have been used as regions of interest (ROI) to calculate, with respect to them, the percentage (%) of surfaces classified as stable during the period 2001–2018 by both the indicator 15.3.1 in the original version (UNCCD LDN) and the improved version (iLDN). In the case of Rome municipality, the iLDN indicator classified correctly as stable all red areas (99.5%), while the UNCCD LDN indicator considered only 66.1% of the red areas as stable.

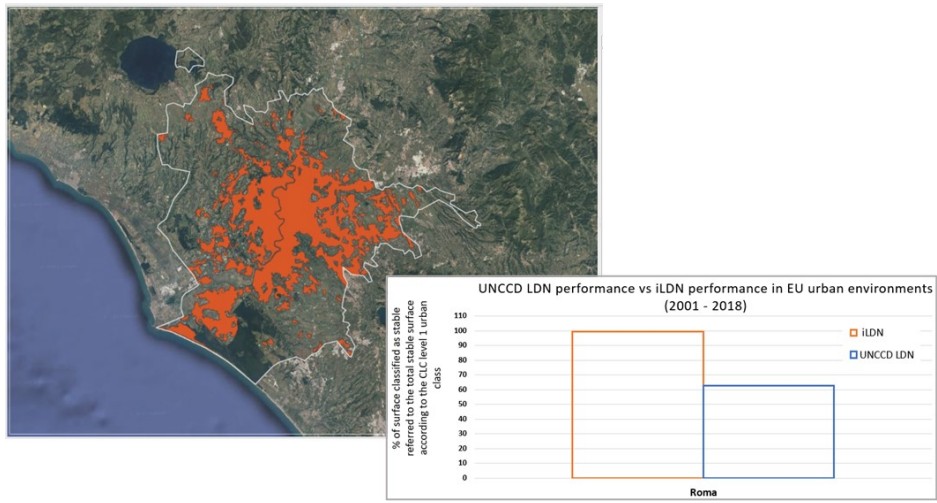

**Figure 5.** The municipality of Rome (white boundaries). Red surfaces represent the urban class (CLC level 1) considered "stable" from 2000 to 2018; The graph shows with respect to red areas, the % of surfaces classified as stable during the period 2001–2018 by both the indicator 15.3.1 in the original version (UNCCD LDN) and the improved version (iLDN).

The procedure described above was applied for the entire EU territory, considering municipalities with a population ranging from 300 K to 3 M inhabitants (source EUROSTAT); The results obtained are reported in Figure 6 where a graph shows, as in the case of Rome, the% of surfaces classified as stable during the period 2001–2018 by both the indicator 15.3.1 in the original version (UNCCD LDN) and the improved version (iLDN); The % are referred to the surfaces falling within the urban class (CLC level 1) considered stable from 2000 to 2018. It is self-evident the better performance of the iLDN indicator, which consistently achieved high percentages of classification considered correct.

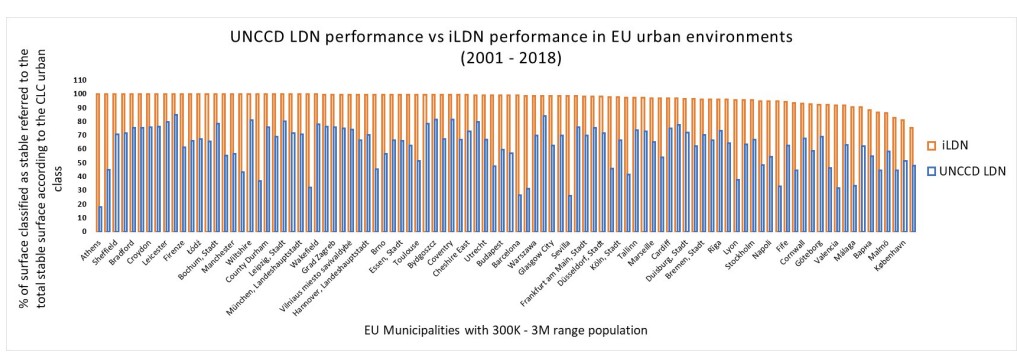

**Figure 6.** The graph shows with respect to the urban areas "stable" (CLC level 1) during the period 2000–2018, the % of surfaces classified as "stable" during the same period by both the indicator 15.3.1 in the original version (UNCCD LDN) and the improved version (iLDN). The data are referred to municipalities with 300 K–3 M range of population (EUROSTAT); iLDN data (orange) are sorted in ascending order.

Figure 7 shows some details of the masking results. On the left, the UNCCD LDN indicator classified large areas within the cities of Rome, Naples, Milan and Berlin as degraded or improved surfaces. Excluding some green urban areas, it is evident that the misclassification is reduced by applying the improving procedure (iLDN), the results of which are visible on the right side.

The improved indicator (iLDN) which actually represents the main layer of the LDN tool was loaded into the SDSS database. In addition, some other raster/vector layers, as described below, complete the dataset of the tool.

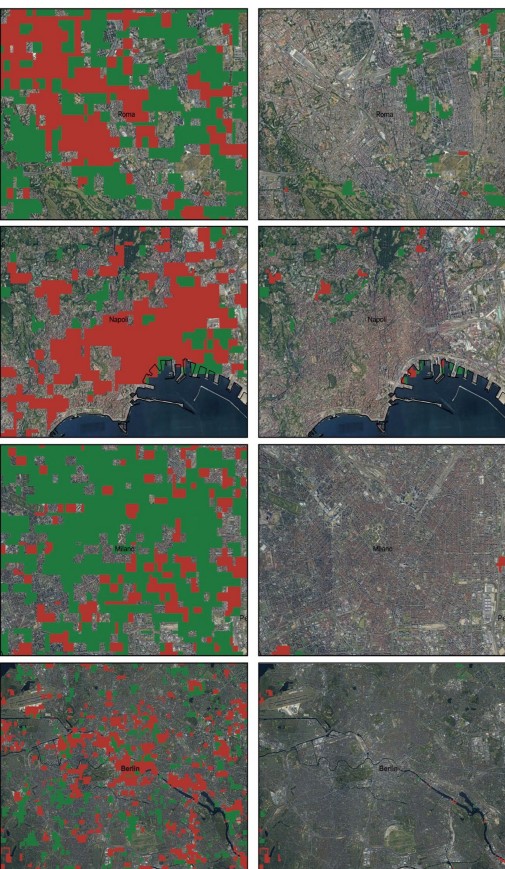

**Figure 7.** On the left, from top to bottom, the UNCCD LDN indicator details for change detection between 2001 and 2018 in the cities of Rome, Naples, Milan and Berlin. On the right, the same cities classified by using the improved indicator (iLDN). Red pixels: areas classified as degraded; green pixels: areas classified as improved; no color: areas classified as stable.

### *3.3. The LDN Tool*

As with all LANDSUPPORT tools, interaction between the users and the system is possible thanks to the free web-based Graphical User Interface (GUI), which includes three sectors: data viewer, map viewer and analysis tools (Figure 8). Data viewer contains the 'layers' sheet, which allows the users to navigate through spatial data levels and the ROIs.

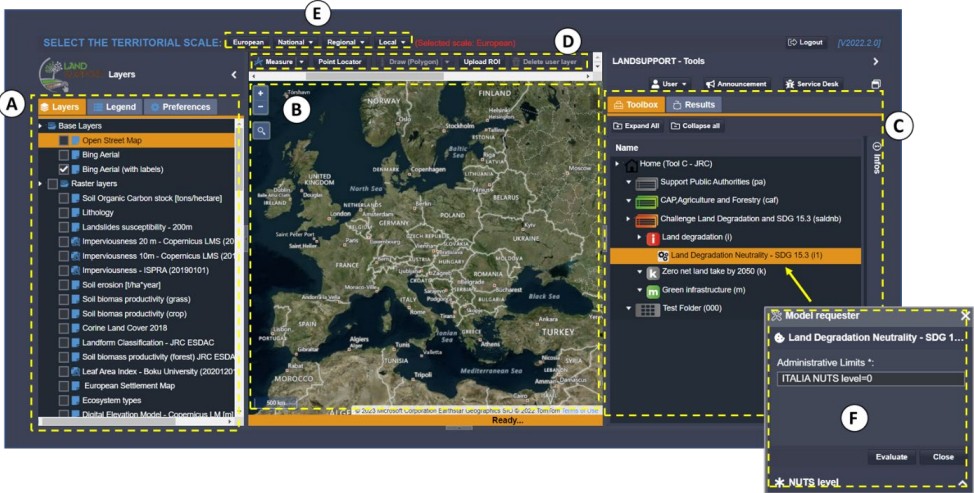

**Figure 8.** Graphic User Interface of the LANDSUPPORT S-DSS. (**A**) Data viewer. (**B**) Map viewer. (**C**) Analysis tool. (**D**) GIS tools. (**E**) Selection of spatial scales. (**F**) Model requester for LDN tool.

The central map viewer displays both the maps selected in the 'layers' sheet and all the maps deriving from tool elaborations.

The analysis tools sector includes the 'toolbox', which allows the users to navigate all of the LANDSUPPORT tools.

In addition to the above, at the very top of the dashboard, there are some typical GIS tools (e.g., draw polygon, measure distances) that enable the users to draw, save and use his ROI for other LANDSUPPORT applications.

The core concept of the LANDSUPPORT system is its capability to process spatial data for the area within the ROI, which, according to the requirements of the specific tool utilized, can be self-drawn by the users or selected from a model requester prompt, such as in the case of LDN tool referring to administrative boundaries.

The LDN tool is accessible at several spatial scales, from the national to municipality level, and the only information it requires from the users is the ROI within which to run the models.

With this consideration, EU administrative levels (from Country to Municipality level) have been added to the database as vector type data to provide boundaries for all the potential administrative ROI the tool may process. These administrative data have been downloaded from Eurostat (https://ec.europa.eu/eurostat/data/database "URL accessed on 5 October 2020"). Figure 6 above shows the right side of the GUI (toolbox) with the pop-up panel for the LDN tool. The users select the ROI (nation, region, province or municipality boundaries) through the panel and runs the tool. The command launches a real time sequence of functions that consist of zonal statistics over the SDG 15.3.1 raster indicator combined with an additional information layer, the Corine land cover raster map for the year 2018, using the ROI as reference to provide end-users with information concerning the main land cover classes affected by degradation or improvement phenomena.

The result consists basically of two types of output: one is a raster map that is clipped by the ROI (i.e., NUT) chosen by the users and shows pixels classified as degraded, improved or stable and the other is a downloadable technical report in PDF format, produced in real time, that contains two groups of data and statistics concerning the rate of change in the degraded, stable or improved surfaces within the ROI. The data are reported in tabular format and expressed both in hectares (ha) and percentages (%) of the total extension of the ROI.

The first information concerns the proportion of degraded, stable or improved land over the total ROI extension. The second group of data provides further detail of this land degradation as it presents the statistics on the main land cover classes (as expressed by Corine land cover map level 1) affected by degradation, remaining stable or improved. Figure 9 shows an example of the output where only the first group of data is reported. In this case, the administrative area chosen as ROI is the entire territory of Italy (NUTS level 0).

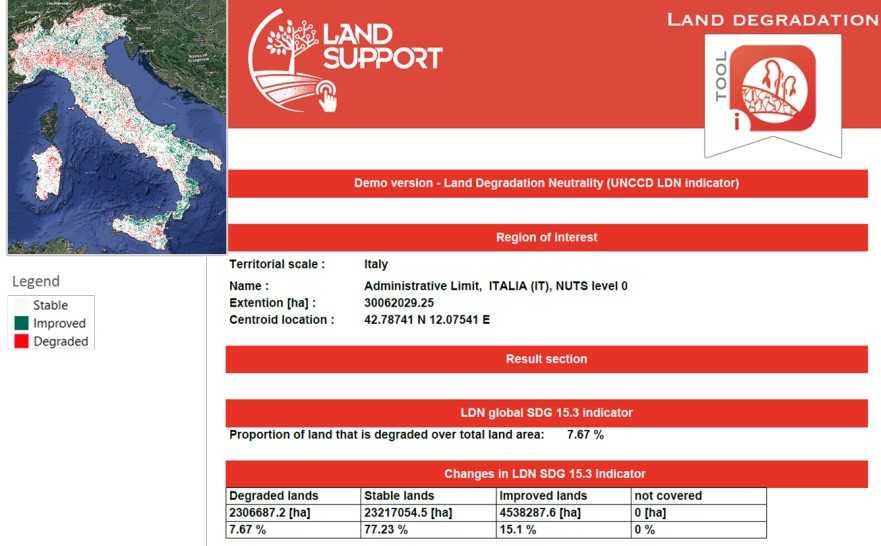

**Figure 9.** Example of the LDN tool output (technical report produced on the fly).

Three overviews of case studies referring to the above application are reported below in order to explore the operation of the LDN tool at different spatial scales.

## 4. Case Studies

### *4.1. Case 1: (National Extension)*

4.1.1. Obtaining the SDG Indicator 15.3.1 at the National Level

In the United Nations 2030 agenda, the soil and land target is to combat desertification, restore degraded land and soil and achieve a land degradation-neutral world through the monitoring of the proportion of land that is degraded over a total area. Here, must be stressed that the evaluation of land degradation by quantifying the SDG 15.3.1 indicator at the national level is mandatory for all EU countries. In addition, targeted interventions at the national level (acting as a reference to regional implementations) would benefit from more detailed analysis, for example by associating the degree of soil degradation to the land use classes most affected.

LANDSUPPORT Tool Help

As a result of the application of on-the-fly GIS processing (e.g., zonal statistics, merging, intersect), the LDN tool provides real time data relating to regions of interest by producing spatial statistics, even on a national scale (NUTS0). Below, Table 1 and the following graphs (Figure 10) report an example of the national-level application for Italy, Hungary, Austria and France.

**Table 1.** Example of zonal statistics by LDN tool applied at the national level.

| | Land Use Class Extension (ha) | | |
|---|---|---|---|
| | **Total ROI Extension** | **Forests** | **Agricultural** |
| Italy | 30,062,029 | 12,428,000 | 15,642,000 |
| Hungary | 1,342,831 | 489,630 | 746,201 |
| Austria | 8,394,780 | 5,127,264 | 2,678,365 |
| France | 63,848,068 | 18,574,106 | 32,388,971 |
| | **Degraded (%)** | | |
| Italy | 7.67 | 8.03 | 7.78 |
| Hungary | 3.56 | 1.52 | 4.94 |
| Austria | 5.64 | 5.59 | 6.06 |
| France | 10.62 | 8.34 | 15.43 |
| | **Improved (%)** | | |
| Italy | 15.10 | 17.17 | 14.45 |
| Hungary | 25.23 | 33.56 | 21.91 |
| Austria | 16.41 | 18.86 | 13.92 |
| France | 11.70 | 20.61 | 10.58 |

According to the statistics on degraded and improved lands, France appears to be in the worst position with almost 11% of its total surface classified as degraded, mainly in agricultural territories (15.43%), and 11.7% of territory classified as improved. Italy has almost 8% of its total surface degraded, distributed equally between forest and agriculture, and 15.1% of lands improved. Austria is the third most degraded (5.64%) with similar levels in forest and agricultural areas and with 16.41% of the surface classified as improved. Finally, Hungary appears to be the least degraded (3.56%), mainly in agricultural areas, and the most improved (25.23%). It is evident from the reported data that the surfaces classified as improved for this case study are greater than those defined as degraded. These "improvement" data are most likely overestimated and it must be accepted with caution

i.e., [10] since it highlights the problem which lies in easy and straightforward application of SDG 15.3.1 criteria as indicated in the introduction.

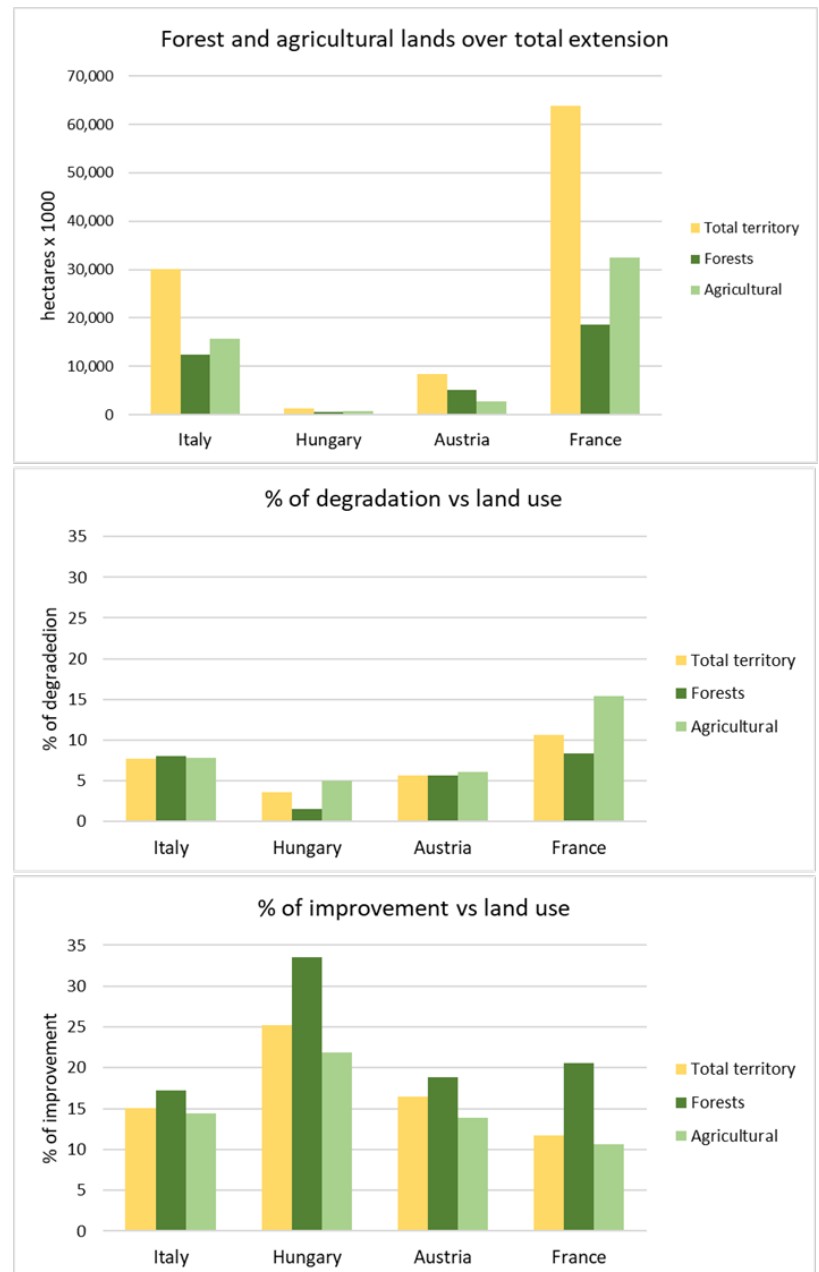

**Figure 10.** Graphic representation of zonal statistics by LDN tool applied at the national level.

In addition, these large improvements may be in contrast with the latest report by the EU Soil Health and Food Mission Board and JRC on the state of health of European soils, which considers approximately 60–70% of EU soils to be degraded [5]. For this specific contribution, this criticality is not crucial since better maps of land degradation can easily be uploaded into the system as soon as they become available. These results show the potential of the LDN tool, which, in just a few seconds, produces data and statistics that are useful for analyzing the state of land degradation within each EU country and enabling comparisons between different geographical areas within the EU. Analysis on a national scale allows a general picture of the degradation or improvement trajectories that the different countries are following as regards their agricultural and forest soils.

*4.2. Case 2: (Regional Extension)*

4.2.1. Improving the Regional Rural Development Plan

In Italy, the Regional Rural Development Plan (also named RRDP) functions as the regional implementation of the National and EU Rural Development Programmes (RDPs) and represents regional interests in setting out priorities so as to address the needs of the specific geographical area it covers. The RRDP is drawn up every five years, but improvements and updates are made annually in line with priorities that are defined with the support of easily available tools which can produce updated spatial statistics on land use/soil conditions over a regional extension.

LANDSUPPORT Tool Help

Case 2 is an example of a procedure employed by a "regional manager" involved in updating the RRDP. In order to achieve this task, he is interested in acquiring information, typically not easily available, on the proportion of degraded lands over the entire regional territory. He is also interested in obtaining a more detailed analysis by identifying the most degraded areas and/or the land cover classes most affected by degradation in the region. This information could constitute a good basis for a general picture of degraded territories over the region and possible identification of areas to focus on with an aim to implement mitigation measures. The regional manager uses the LDN tool and chooses the entire Campania territory (south-west Italy) as his/her ROI. The output obtained is the PDF report with data and statistics on the proportion of degraded, stable or improved surfaces within the region and a map of these surfaces. Basically, he/she now knows the rate of degradation, where it has been detected and which land cover classes are most affected. In Figures 11 and 12 below, the results are shown, respectively, of the SDG 15.3.1 indicator superimposed over the territory of Campania region, the report with zonal statistics indicating the magnitude of degraded/improved lands across the territory and data aggregated by land use classes.

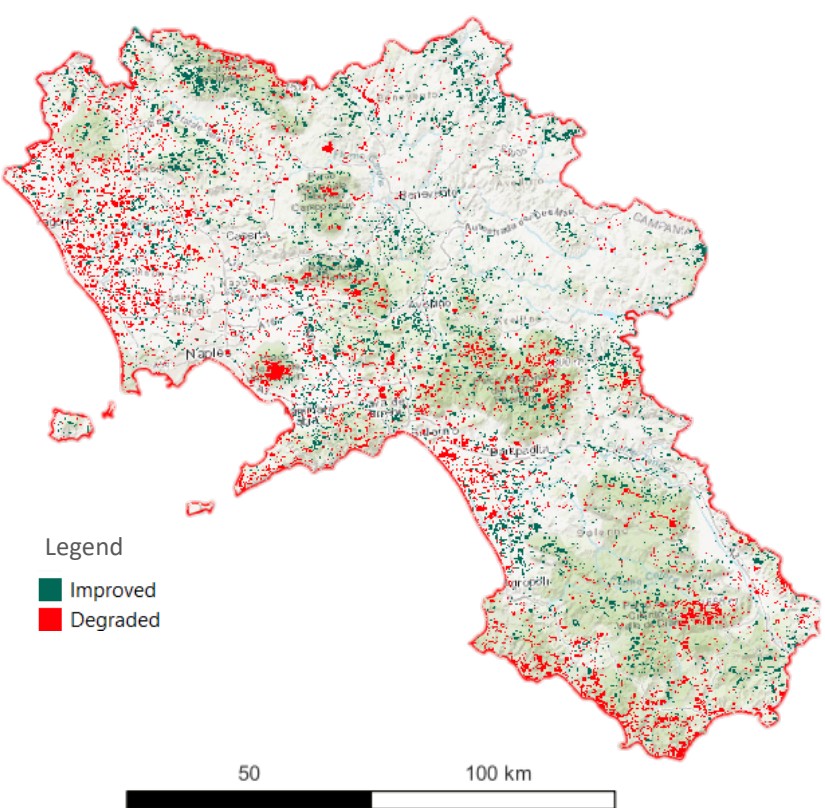

**Figure 11.** Results of the SDG 15.3.1 indicator superimposed over the territory of Campania.

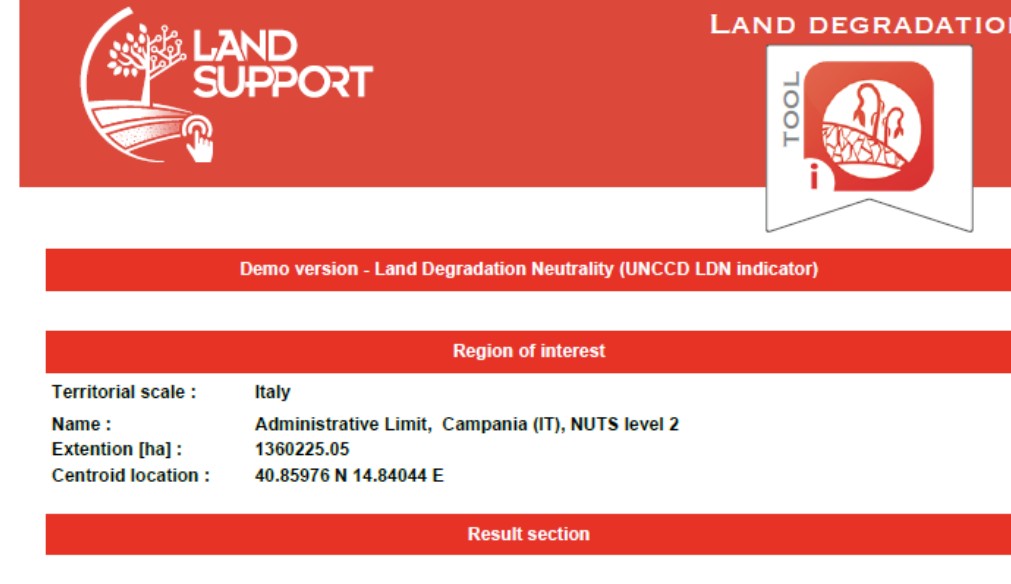

**Figure 12.** Zonal statistics indicating the magnitude of degraded/improved lands across the territory and the data aggregated by land use classes.

The results show degraded lands (red) for 5.71% of the territory, which is quite similar to the improved (green) surfaces (5.48%), while the remaining surface is classified as stable (no color). When observing the map, it is evident that the degraded lands are mostly concentrated along the coasts and on the plains where intensive agriculture is a common practice. Most of these concentrations are due (few of these data were checked in the field) to the installation of greenhouses and new urbanization, as well as to change in land use and management resulting in possible soil erosion processes. In the middle west of the map, there is a degraded spot corresponding to the Vesuvius volcano; this is probably related to large fires which occurred during the summer of 2017. Some other spots are located in hilly and mountainous areas along the Apennine chain (south east—north west) where

the cause might be attributable to the management of coppices (mainly chestnuts), which appears as sudden drops in the NDVI, or the effects of drought periods which cause severe water stress for plants. The improved lands are more dispersed, mostly in the inland hilly areas. This is possibly due to areas of natural forest which generally show an increasing NDVI, while the improvements recorded in the plain areas are mostly due to changes in land use and management.

All this information, delivered on-the-fly by the LDN tool, constitutes a clear synoptic picture of the current territorial condition which can be used by the regional manager to obtain a more detailed analysis and plan effective measures for degradation containment. In this specific case, such situations occur in plain areas, where large-scale soil sealing occurs, and in inland areas, where inadequate forest management is linked to soil degradation phenomena such as compaction and erosion.

### 4.3. Case 3: (Municipalities)

### 4.3.1. Updating the Municipal Urbanization Plan (PUC) of the Naples Municipality

Changes in land use in the Italian municipalities are supervised through development, implementation and periodic updates of the so called PUC (Municipal Urbanization Plan, an update on the General Regulatory Plan that was established by Italian Law 1150/1942) and which consists of maps, technical documents and regulations. The Plan is a prerogative of urban planners, who are assisted by other experts such as pedologists, geologists, agronomists and lawyers. The document is periodically reviewed due to the great influence it has on the territory through its (i) protection, enhancement and creation of green areas and urban parks, (ii) definition of and rules regarding land use, (iii) delineation of different zones with different uses within the Municipal territory.

LANDSUPPORT Tool Help

In this context, the LDN tool is appropriate for use in peri-urban areas and urban green areas as it provides a spatial assessment of the proportion of land likely to be affected by soil degradation/improvement within the municipality and the most affected land use categories. Case 3 shows application of the tool by an urban planner with an intention to enrich the PUC with data regarding the presence of degraded zones within the municipality of Naples (Campania region, Italy) and outline areas to be reported as threatened in terms of potential crop productivity, urban forest/park health and human well-being. The identification of areas of land degradation can also potentially affect the revision of urban zoning (classes A, B, C, D and E), so enabling a better definition of areas of urban expansion, for instance (C zones), or the planning of action to recover agriculture areas (E zones).

In our specific case, the planner uses the LDN tool by choosing the municipality of Naples (NUT level 4) as his/her ROI. The output obtained is the PDF report with data and statistics on the proportion of degraded, stable or improved surfaces within the municipality and a map of these surfaces indicating where they have been identified. In Figure 13 and Table 2 below, the indicator map for the municipality of Naples and the zonal statistics on the state of degraded/improved surfaces within the ROI are reported.

The indicator classified approximately 187 hectares of land as degraded and 259 as improved, which were, respectively, 1.7% and 2.3% of the total area of the Municipality. The remaining area is classified as stable. Degraded surfaces are spread in an arc from east to west of the municipality through the less dense and more fragmented urban zones intermingled with green areas. Most likely the new degradation is the result of further widespread urbanization during the monitored period. The areas classified as improved follow a similar trend with a concentration in the northwest corresponding to a protected nature park, whose vegetation is evidently in a growth phase. With these results, the expert users have a picture of the condition of the territory according to the indicator and can draw initial conclusions for the updating of the PUC on: (i) peri-urban areas of the municipality where agricultural environments intermingled with urban areas are undergoing a process of widespread degradation relating to a loss of soil and its functions; (ii) wooded areas

within the municipality, green environments to be protected that are available to citizens, show positive trends or are classified as stable, and therefore are not negatively altered.

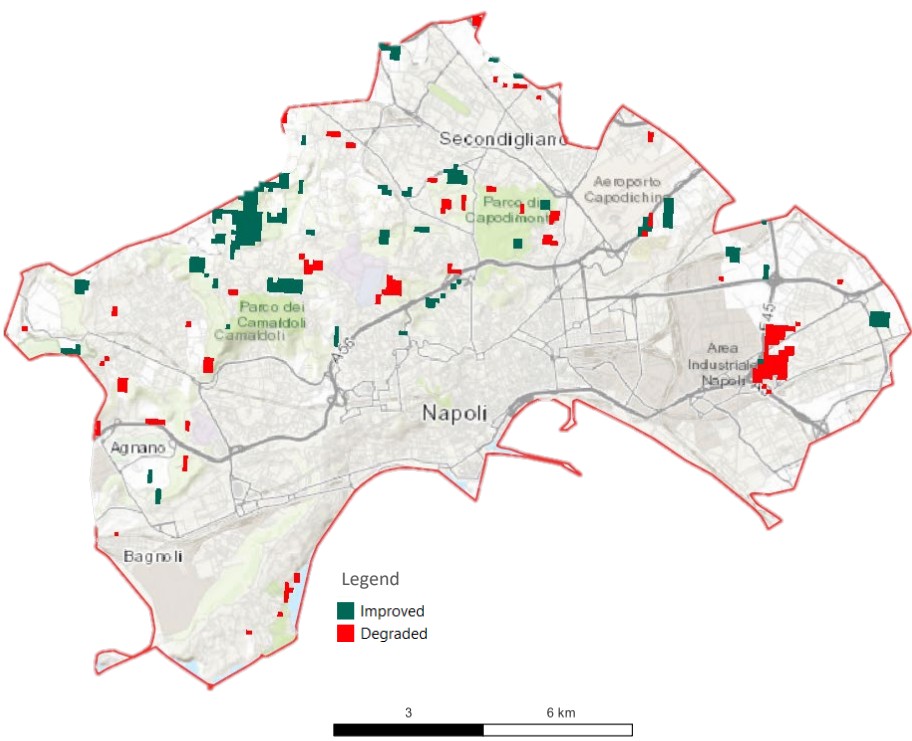

**Figure 13.** Surfaces classified as degraded or improved according to the SDG indicator for the municipality of Naples.

**Table 2.** Zonal statistics representing the proportion of degraded/improved surfaces within the selected ROI.

| Indicator Class | Surface Affected (ha) | % Referred to Total ROI Area | Total ROI Area |
|---|---|---|---|
| Degradation | 186.9 | 1.7 | |
| Improvement | 259.1 | 2.3 | **11,131.2 ha** |
| Stable | 10,685.2 | 95.9 | |

This case study gives us the opportunity to stress how important it would be especially at the local scale, to calculate and eventually validate the indicator using as base data high-resolution and/or additional information (e.g., local dataset, LUCAS dataset). At the same time it would be crucial to both include with greater weight in the calculation, recent changes in land cover (e.g., recent urbanization) and also consider a local set of productivity thresholds to address individual land threats as suggested by [20]. In such cases (e.g., urban and peri-urban areas with urbanization intermixed with green areas), new data and rules could significantly change the results of the computations by highlighting local issues that currently are overlooked.

## 5. Discussion

In this paper, the importance of a Spatial Decision Support system built through Geospatial Cyberinfrastructure in supporting LDN goals from EU to municipality level has been highlighted. This system potentially allows users (expert and not) to evaluate the soil degradation in their region and integrate other connected environmental themes. The great potential of this approach through an operational system which is found at

www.landsupport.eu has been demonstrated. The system is flexible and open to further implementation of both models and databases. It is, therefore, important to discuss some key points of the approach and the lessons learned, as well as to highlight important problems.

The LDN tool effectively provides the knowledge base for end-users to understand the soil degradation situation in specific territories and allows a multilevel comparison between different territories, in order to improve awareness among stakeholders and ordinary citizens who want reliable and comparable data on the subject. However, it is necessary to point out that in its current form the SDG 15.3.1 indicator has greater applicability at the territorial scale than at the local scale. This emerges clearly for example from our analyses in urbanized environments, where we realized that green areas intermixed with sealed surfaces may not be "seen" by the indicator when sparse and scattered, or may cause misclassifications due to edge effect when large and more dense. This is clearly an issue of spatial resolution of the indicator components. Not surprisingly, results for the SDG 15.3.1 indicator describe a land degradation situation in Europe that needs to be investigated in finer thematic detail and geometric resolution. One solution to these issues could be to use different data, with different spatial resolution, to calculate the indicator at different scales. For example, application at the local scale could include the use of Landsat or Sentinel data for the calculation of productivity metrics.

However, in the meantime, this work sets the basis for some methodological improvements and comparison. The set of three sub-indicators proves to be a flexible, valid and homogeneous starting point. Nevertheless, major limitations must be considered, including the difficulty of obtaining an up-to-date database on land cover and SOC data and the excessive sensitivity of the model to NDVI changes due to the current thresholds of the Trends.Earth methodology. The last stress the need to consider new rules in the indicator calculation such as add a specific set of thresholds for local productivity.

Indeed, to start with, these problems may lead to an underestimation of the SOC and productivity variations which can occur with an irregular spatial distribution that makes them only detectable through in situ sampling or very high-resolution multispectral image classification. Second, the sensitivity of the NDVI model leads to an overestimation of the areas that show improvement) or degradation (especially in denuded areas), due to temporary changes in vegetation coverage linked to temporary periods of abundant rainfall or drought.

Indeed, our first reliability check on the indicator was performed by observing the results for dense urban areas where there was certainty that the surfaces had remained stable, with no land degradation or improvement, over time. The approach revealed artefacts that were misclassifications of these surfaces, which were most probably classified as improved or degraded due to a mix of poor spatial resolution (it is probable that some small urban green areas affected the signal) and the sensitivity of the NDVI model. On the basis of this evidence, the land degradation map used for the LDN tool was corrected so as to avoid misclassification in urban areas. This improvement does not, though, solve all the overestimation issues of for land improvement in rural areas.

However, despite these problems and especially for large territories, the indicator has to be considered a valuable instrument for identifying areas in which priority actions are needed in order to achieve LDN.

## 6. Conclusions

In this paper, the development and usability of a tool designed to deliver free web-based operational support for a very large territory on several spatial levels have been described. We believe that such tools are to be considered fundamental instruments in increasing awareness of the topic.

Some technical improvements for the methodology of LDN calculation are still needed and the use of NDVI as a proxy indicator of net primary productivity is still too unreliable, both for the resolution and the current thresholds for degradation or improvement detection. Some challenges, such as climatic correction and integration with other sub-indicators, still

need to be addressed. On the other hand, the concern about the availability of multiscale spatial and temporal data could be solved in the near future, at least for Europe, given the increasing availability of Copernicus products on land cover and Land Productivity. However, these will not cover current reporting requirements. In this sense, an effort to homogenize data from different sources will be necessary, just as will be new methods of soil organic carbon estimation through remote sensing, which might solve the availability and updatability issues this sub-indicator has.

The tool has proved itself effective in identifying where it is most crucial to focus efforts on land restoration practices, but it could still be integrated with more detailed tools to increase accuracy and define the main causes of degradation.

In this paper, the tool has been proven also through a short selection of practical case studies, where data, tables and stats are provided to the users aiming to challenge land degradation at different spatial extents.

Future improved versions of the indicator and/or additional indicators will be easily loaded into the LDN tool database and made available to users.

**Author Contributions:** Conceptualization, P.M.; methodology, P.M. and F.A.M.; software, G.L.; validation, M.D.L., N.R. and M.M.; resources, M.D.L., M.M. and N.R.; data curation, P.M. and F.A.M.; writing—original draft preparation, P.M.; writing—review and editing, A.A., M.B., F.A.M., G.F., M.M., M.D.L. and N.R. All authors have read and agreed to the published version of the manuscript.

**Funding:** This paper refers to the LANDSUPPORT project which has been funded within EC Horizon 2020, grant number: 774234.

**Data Availability Statement:** The data that support the findings of this study are available on request from the corresponding author. The data are not publicly available due to privacy or ethical restrictions.

**Acknowledgments:** This paper was produced within the framework of the LANDSUPPORT projects (Horizon 2020, 774234). The authors would like to thank Luciana Minieri and Michele Flammia for their contribution to support project activities and Roberto De Mascellis and Nadia Orefice for their contribution to research activities.

**Conflicts of Interest:** The authors declare that they have no known competing financial interests or personal relationships that could have appeared to influence the work reported in this paper.

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
