# Peer review of "A Geospatial Decision Support System for Supporting the Assessment of Land Degradation in Europe"

_land, doi:10.3390/land13010089_

Round 1

Reviewer 1 Report

Comments and Suggestions for Authors

Dear authors,

I thoroughly enjoyed reading your article. I have only two comments: 

1. please consider whether to drop the term 'we' from the text. Perhaps it would be better to have the whole text written impersonally. 

2. while reading the article, part 3 'results and discussion' confused me. This section appears right after materials and methods, it seems that this is not the right place for this section or the name of this section is not correct. Especially as section 5 is also called discussion. I suggest changing the title of section 3, perhaps to 'research'.

Author Response

Dear reviewer, thank you for the evaluation of our work. We will consider your comments in their whole. Below our comments to your review:

  1. please consider whether to drop the term 'we' from the text. Perhaps it would be better to have the whole text written impersonally. 

Thank you. The term ‘we’ has been dropped.

  1. while reading the article, part 3 'results and discussion' confused me. This section appears right after materials and methods, it seems that this is not the right place for this section or the name of this section is not correct. Especially as section 5 is also called discussion. I suggest changing the title of section 3, perhaps to 'research'.

Thank you. The title of section 3 is now “Results”

Reviewer 2 Report

Comments and Suggestions for Authors

In this paper authors aim to describe a Geospatial Decision Support System for Supporting the Assessment of Land Degradation in Europe, based on the application of the SDG 15.3.1 indicator and the Trends.Earth software. Their goal and purpose is relevant and current for respective scientific communities but the paper presentation lacks of clarity and sufficient details, both in technical and structural regards, but also on how the presented tool and achieved results answer the problems stated in Introduction paragraph, L70-89.

General Comments

1. The main contribution of this paper is the development of an operational LDN tool to visualize the SDG 15.3.1 indicator at NUTS level for the entire EU territory. Throughout the manuscript it should be clear that the LDN tool is used to clip the, at EU-level calculated indicator, at the specific ROI and not to produce the indicator at the specific regional ROI-specified level. The new feature that is provided in this tool is based on the visualization and reporting of results at the NUT level .Please be clear in regards to that throughout your text.

2. In relation to the previous comment, how the authors comment and discuss on the fact that this tool is suggested and presented as suitable for different spatial scales, but the scalability refers only to the visualization, since the 15.3.1 indicator is computed at the EU-level? Some relevant issues in regards to that:

2a. The Land Productivity sub-indicator is based on spatial comparison and results may change if the extend over which the analysis is conducted changes (Sims 2021, https://www.unccd.int/sites/default/files/relevant-links/2021-03/Indicator_15.3.1_GPG_v2_29Mar_Advanced-version.pdf ).

2b. How can degradation and improvement be homogeneously defined at the EU level and for different applications and land cover problems.

3. The second contribution of this paper is claimed to be a reliability assessment of the Trends.Earth approach followed by a potential improvement in the SDG 15.3.1 indicator. The reliability assessment is rather limited, since it is based on qualitative observations of specific urban areas, with no quantitative indications, or indications on how many or how large were the areas tested. Please be more clear and elaborate more on this.

4. The authors should in general re-examine the structure of the paper, what parts should be included in what respective sections, e.g. there is no matching between methods and results, no methodololy provided for the LDN. In general better connection between the sections, sub-section and paragraphs should be achieved, providing more, and more detailed information for all parts.

Specific comments.

1.      Abstract: The abstract should be more indicative of the work that has been done. There is no reference on the SDG 15.3.1 indicator and in L23-27, there is much talk for the Landsupport platform, that can be a bit misleading in regards to what is being presented on the current paper.

2.      L31. systems?

3.      L33, L81, etc. Explain abbreviations 1st in manuscript.

4.      L108. Check double/triple spaces all over the text.

5.      L116-118. Please refer and include in the introduction the respective and relevant visualizations systems and their characteristics.

6.      L121. Automatic link does not work.

7.      L177-178. How you have dealt with detected anomalies?

8.      L185,L588. There is a version 2.0 published of the GPG  (Sims 2021, https://www.unccd.int/sites/default/files/relevant-links/2021-03/Indicator_15.3.1_GPG_v2_29Mar_Advanced-version.pdf )

9.      L196-198, L220-222. How about bush encroachment cases in southern Europe due to increased aridity?

10.   L210-211. Please provide the exact correspondence between CLC classes and the land cover classes used in your application.

11.   L228-229. How you have dealt with detected anomalies?

12.   Figure 3/4: There is no indication of the spatial location of the images.

13.   L269. With regard to all

14.   L277-278. Why doi and not #ref?

15.   L279-280. What about improvements in terms of expansion of urban green throughout the years that were not reported due to this assumption and exaggeration? Elaborate more here and on the discussion.

16.   L454-457 Case study 3. Again in regards to comment 14, how can the the LDN tool be considered as appropriate for use in peri-urban areas and urban green areas? Especially for the latter, the CLC class at the used level of aggregation does not seem to define a separate class.

17.   Figure 7/10. What is denoted with “no covered” from the Landsupport statistics.

18.   L500-503. Please be careful and clear in regards to general comment 1.

19.   L504-505. Please elaborate more, providing also a discussion in regards to at least ref [20] https://onlinelibrary.wiley.com/doi/10.1002/ldr.4457 , e.g., Evaluation of the United Nations Sustainable Development Goal 15.3.1 indicator of land degradation in the European Union, they propose identification of high risk areas and conduction of case studies with HR Data and setting of local thresholds.

20.   L506. How does it set the basis?

21.   L515-520. Please rephrase, its not clear.

22.   L525-526. “it is probable that some small urban green areas affected the signal”. Could some small urban green areas have been created as well?

Comments on the Quality of English Language

Minor editing of English language required through proof-reading.

Author Response

Dear reviewer, thank you for your comments which we will try to consider in their whole. Below our comments to your review point by point:

  • The reviewer states that our work is not clear in the following: "how the presented tool and achieved results answer the problems stated in Introduction paragraph, L70-89".

Actually the work we have presented does not  aspire to answer all the issues identified, but rather aims to make a contribution to an improvement of the current official indicator for land degradation, and wants to present the potential that such an indicator can have when integrated into a decision support system, as reported between L107-118;

  • 1. The main contribution of this paper is the development of an operational LDN tool to visualize the SDG 15.3.1 indicator at NUTS level for the entire EU territory. Throughout the manuscript it should be clear that the LDN tool is used to clip the, at EU-level calculated indicator, at the specific ROI and not to produce the indicator at the specific regional ROI-specified level. The new feature that is provided in this tool is based on the visualization and reporting of results at the NUT level .Please be clear in regards to that throughout your text. 

Here it is necessary to emphasize that the tool presented here does not just clip ROIs on the degradation indicator layer, but produces also statistics by cross-referencing the indicator classes with other (potentially many others if required) data. In the specific case, land cover data from Corine products. Regarding clarity on how the tool works and its applicability, we quote here the points in the paper where what is required is specified: L159-168 where it is clarified that the tool is based on specific models; L324-348 where it is clarified that the main concept of the system is that the ROIs in which data processing takes place can be selected by the user as in the case of administrative boundaries;  L318-320 where it is stated "With this consideration, EU administrative levels (from Country to Municipality level) have been added to the database as vector type data to provide boundaries for all the potential administrative ROIs the tool may process; L323-328 where it is stated that "The user selects the ROI (nation, region, province or municipality boundaries) through the panel and runs the tool. The command launches a real time sequence of functions that consist of zonal statistics over the SDG 15.3.1 raster indicator combined with an additional information layer, the Corine Land Cover raster map for the year 2018, using the ROI as reference to provide end-users with information concerning the main land cover classes affected by degradation or improvement phenomena"; L329 - 331 where it is reported "The result consists basically of  two types of output: one is a raster map that is clipped by the ROI chosen by the user and shows  pixels classified as degraded, improved or stable and the other is a downloadable technical report in PDF format, produced in real time...;

However, for clarity we improved some sentences as follow:

L314-317 (original): “The core concept of the LANDSUPPORT system is its capability to process spatial data for the area within the ROI, which, according to the requirements of the specific tool utilized, can be self-drawn by the user or selected from a model requester prompt, such as in the case of referring to administrative boundaries.

L327-330 (reviewed): “The core concept of the LANDSUPPORT system is its capability to process spatial data for the area within the ROI, which, according to the requirements of the specific tool utilized, can be self-drawn by the user or selected from a model requester prompt, such as in the case of LDN tool referring to administrative boundaries.”

L335 – 336 (original): “The result consists basically of two types of output: one is a raster map that is clipped by the ROI chosen by the user”

L348 – 349 (reviewed): “The result consists basically of two types of output: one is a raster map that is clipped by the ROI (i.e. NUT) chosen by the user”

  • 2. How the authors comment and discuss on the fact that this tool is suggested and presented as suitable for different spatial scales, but the scalability refers only to the visualization, since the 15.3.1 indicator is computed at the EU-level?

We agree with the reviewer. Here we would to specify that we consider that the SDG 15.3.1 indicator is produced with a coverage of the European territory and its spatial resolution (Productivity sub-indicator based on MODIS 250m data), makes it applicable up to the scale of the administrative limit. On the other hand, between L530-551 in the conclusions paragraph we address this issue.

  • 2a. The Land Productivity sub-indicator is based on spatial comparison and results may change if the extend over which the analysis is conducted changes (Sims 2021, https://www.unccd.int/sites/default/files/relevant-links/2021-03/Indicator_15.3.1_GPG_v2_29Mar_Advanced-version.pdf ).

The comment is very interesting. In our experience we verified that in attributing the degradation class to the pixel scale, the use of the "one out all out" approach and the logic in the combination of the metrics for the productivity sub-indicator, tend to emphasize the weight of the "trend" metric" which is not related to spatial comparison. Therefore, we considered negligible the weight of the "performance" metric, which is instead linked to the spatial comparison.

  • 2b. How can degradation and improvement be homogeneously defined at the EU level and for different applications and land cover problems.

The comment contains numerous issues which are currently the subject of research throughout Europe. However, in the work we presented we did not have the objective of proposing our own indicator for the definition of degraded areas. We wanted to make an improvement to an indicator currently recognized at European level and we wanted to implement this "improved" indicator in a DSS that could make it more easily applicable and interpretable. Taking into account what is stated above, namely that in most cases it is the "trend" metric that defines whether an area is degraded or not.

  1. The second contribution of this paper is claimed to be a reliability assessment of the Trends.Earth approach followed by a potential improvement in the SDG 15.3.1 indicator. The reliability assessment is rather limited, since it is based on qualitative observations of specific urban areas, with no quantitative indications, or indications on how many or how large were the areas tested. Please be more clear and elaborate more on this.

Thank you for this comment. We checked and corrected the sentence al L113-115 as follow:

Original: : (i) a reliability assessment of the Trends.Earth, followed by a potential improvement in the SDG 15.3.1 indicator

Reviewed: : (i) a reliability assessment of the Trends.Earth approach limited to urbanized areas, followed by a potential improvement in the SDG 15.3.1 indicator

Moreover, we added additional materials (appendix A) where we report the results of a quantitative evaluation of the indicator reliability assessment in EU urban environments. The approach shows and compare the indicator performance in its original and improved version using as reference data the urban surfaces classified as stable (during the period 2001 – 2018) by the Corine Land Cover products. The results show (quantitatively) how the masking procedure improved the indicator reliability in urban environments.        

  1. The authors should in general re-examine the structure of the paper, what parts should be included in what respective sections, e.g. there is no matching between methods and results, no methodology provided for the LDN. In general better connection between the sections, sub-section and paragraphs should be achieved, providing more, and more detailed information for all parts.

The comment is rather general not easy to interpret. With reference to the methodology applied in the development of the LDN tool, we have included a new reference L122 [22] to a very recent publication dedicated to the LANDSUPPORT platform and the logic behind development of the SDSS and all the integrated tools. This manuscript follows the general structure of previous works dedicated to other SDSS tools published in various journals. We therefore opted to maintain this structure.

  • Specific comments.
  • 1.      Abstract: The abstract should be more indicative of the work that has been done. There is no reference on the SDG 15.3.1 indicator and in L23-27, there is much talk for the Landsupport platform, that can be a bit misleading in regards to what is being presented on the current paper.

Thank you for this comment. We modified the text as shown below where you find the old version (Abstract) and the new version (REV-Abstract)

Abstract:      Nowadays, Land Degradation Neutrality (LDN) is on the political agenda as one of the main objectives in order to respond to the increasing degradation processes affecting soils and territories. Nevertheless, proper implementation of environmental policies is very difficult due to a lack of the operational, reliable and easily usable tools necessary to support political decisions when identifying problems, defining the causes of degradation and helping to find possible solutions. It is within this framework that this paper attempts to demonstrate that a new type of Spatial Decision Support System (S-DSS) that is developed on a Geospatial Cyberinfrastructure (GCI) might provide a valuable web-based operational tool which could be offered to EU administrative units (e.g. municipalities) so that they may better evaluate the state and the impact of land degradation in their territories. The land degradation data utilized were obtained from a platform named Trends.Earth, designed to monitor land change by using earth observations, and post-processed to correct some of the major artefacts relating to urban areas.  The S-DSS (www.landsupport.eu) has also been designed to encourage use by multi-user commu-nities (from citizens to scholars, associations and public bodies). Moreover, it supports the acquisi-tion, management and processing of both static and dynamic data, together with data visualization and computer on-the-fly applications, in order to perform modelling, all of which is potentially accessible via the Web. The Land Degradation tool, is designed to support land planning and management by producing data, statistics, reports and maps for any EU area of interest. The tool will be demonstrated through a short selection of practical case studies where data, table and stats are provided to challenge land degradation at different spatial extents. Currently there are WEBGIS system to visualise land degradation maps but – to our knowledge – this is the first SDSS tool enabling a customized LDN reporting at any NUTS level for the entire EU territory.

REV-Abstract: Nowadays, Land Degradation Neutrality (LDN) is on the political agenda as one of the main objectives in order to respond to the increasing degradation processes affecting soils and territories. Nevertheless, proper implementation of environmental policies is very difficult due to a lack of the operational, reliable and easily usable tools necessary to support political decisions when identifying problems, defining the causes of degradation and helping to find possible solutions. It is within this framework that this paper attempts to demonstrate that a new type of Spatial Decision Support System (S-DSS) that is developed on a Geospatial Cyberinfrastructure (GCI) might provide a valuable web-based operational LDN tool which could be offered to EU administrative units (e.g. municipalities) so that they may better evaluate the state and the impact of land degradation in their territories. The S-DSS supports the acquisition, management and processing of both static and dynamic data, together with data visualization and computer on-the-fly applications, in order to perform modelling, all of which is potentially accessible via the Web. The land degradation data utilized to develop the LDN tool refer to the SDG 15.3.1 indicator and were obtained from a platform named Trends.Earth, designed to monitor land change by using earth observations, and post-processed to correct some of the major artefacts relating to urban areas.  The  tool, is designed to support land planning and management by producing data, statistics, reports and maps for any EU area of interest. The tool will be demonstrated through a short selection of practical case studies where data, table and stats are provided to challenge land degradation at different spatial extents. Currently there are WEBGIS system to visualise land degradation maps but – to our knowledge – this is the first S-DSS tool enabling a customized LDN reporting at any NUTS level for the entire EU territory.

  • 2.      L31. systems?

Corrected

  • 3.      L33, L81, etc. Explain abbreviations 1st in manuscript.

L33: S-DSS is explained in L18; LDN is explained in L12; NUTS added “(nomenclature of territorial units for statistics)”

L81: added "(Soil Organic Carbon stock)”

  • 4.      L108. Check double/triple spaces all over the text.

Thank you for the comment. It seems to be a problem generated by the formatting done by the journal.

  • 5.      L116-118. Please refer and include in the introduction the respective and relevant visualizations systems and their characteristics.

We add the sentence: “such as LandGis (https://opengeohub.org/article/pre-release-landgis/) presented as the “OpenStreetMap for land-related environmental data”, Queensland soil and land re-source data web map service (https://www.data.qld.gov.au/dataset/queensland-soil-and-land-resource-data-web-map-service) which contains soil site data and soil polygon mapping data which is rep-resented as four main types: soil, land management and land degradation mapping, Land Portal (https://landportal.org/book/indicator/un-aglnddgrd) expressly dedicated to several land indicators including the SDG 15.3.1”

  • 6.      L121. Automatic link does not work.

Thank you. We checked the system, currently it works

  • 7.      L177-178. How you have dealt with detected anomalies?

In the specific case reported in the text we were referring to problems related to the polygons of the administrative boundaries of different NUTS, that were pre-processed in the postgreSQL environment. We preferred to avoid details of the procedures applied in order to not burden the text.

  • 8.      L185,L588. There is a version 2.0 published of the GPG  (Sims 2021, https://www.unccd.int/sites/default/files/relevant-links/2021-03/Indicator_15.3.1_GPG_v2_29Mar_Advanced-version.pdf )

Thank you, we checked and replaced

  • 9.      L196-198, L220-222. How about bush encroachment cases in southern Europe due to increased aridity?

Thank you for the issue. The one raised is a problem that could probably be mitigated by using higher spatial resolution data in calculating NDVI trajectories and integrating additional vegetation indicators. For example, in the future, use Landsat or Sentinel data to calculate productivity metrics.

  • 10.   L210-211. Please provide the exact correspondence between CLC classes and the land cover classes used in your application.

Thank you for the comment. We checked and replaced.

  • 11.   L228-229. How you have dealt with detected anomalies?

We detailed how we dealt with this issue in chapter 3.1. Moreover, we added more details by responding to the question about the “reliability assessment”

  • 12.   Figure 3/4: There is no indication of the spatial location of the images.

The figures to which the reviewer refers are only examples with the aim of observing some details, so we did not consider important to report the coordinates. In one case we observe how indicator 15.3.1 classifies evident (randomly chosen) cases of degradation linked to urbanization, and in the other case how the GEE code constructs the trajectories for specific pixels. Image 4 refers to the city of Naples.

  • 13.   L269. With regard to all

Corrected

  • 14.   L277-278. Why doi and not #ref?

Corrected

  • 15.   L279-280. What about improvements in terms of expansion of urban green throughout the years that were not reported due to this assumption and exaggeration? Elaborate more here and on the discussion.

The procedure we applied to improve the indicator involved masking urban areas only in the portion that remained stable over the years according to the CORINE dataset. This implies that the masking mainly involved the historical centers of urban areas and the compact high-density urban areas in which the evolution of green areas can be considered a negligible phenomenon, especially when considering the resolution of the MODIS data used to calculate productivity metrics. Here we want to emphasize that our approach does not claim to make perfect the indicator 15.3.1, but rather to improve it. In the supplementary materials this improvement has been quantified.

However, following the reviewer comment we have made the following changes between L288-293:

Original: We acknowledge that CLC1s for urban areas – used as a measure of soil sealing  – typically produce an overestimation (e.g. green infrastructure within the urban centre is considered urban) [32].

Reviewed: . Even considering that this procedure covered compact urban areas and historic city centers, which at the spatial scale of the indicator are actually almost totally sealed surfaces, we acknowledge that CLC1s for urban areas – used as a measure of soil sealing  – typically produce an overestimation (e.g. green infrastructure within the urban centre is considered urban) [32].

  • 16.   L454-457 Case study 3. Again in regards to comment 14, how can the the LDN tool be considered as appropriate for use in peri-urban areas and urban green areas? Especially for the latter, the CLC class at the used level of aggregation does not seem to define a separate class.

As reported for comment 15, the LDN tool has been improved only in very compact and stable urban areas; in peri-urban and urban green areas still works the original version of the indicator. Here the presence of improved, stable or degraded surfaces depends from the sensitivity of the approach (i.e. the trajectory metric based on NDVI which has higher weight in the final class attribution).

  • 17.   Figure 7/10. What is denoted with “no covered” from the Landsupport statistics.

“No covered” is a field added to justify surfaces not involved in the statistics due to the clipping process which overlap ROI polygons (vector layer) with a raster at 250m pixel resolution. This procedure could exclude some pixels along the edges of the ROI because they fall less than 50% inside the polygon.

  • 18.   L500-503. Please be careful and clear in regards to general comment 1.

Thank you for the comment. We followed the suggestion changing the sentence as follow:

Original: i)protection and enhancement of natural landscape and cultural heritage;

Reviewed: i) protection, enhancement and creation of green areas and urban parks

  • 19.   L504-505. Please elaborate more, providing also a discussion in regards to at least ref [20] https://onlinelibrary.wiley.com/doi/10.1002/ldr.4457 , e.g., Evaluation of the United Nations Sustainable Development Goal 15.3.1 indicator of land degradation in the European Union, they propose identification of high risk areas and conduction of case studies with HR Data and setting of local thresholds.

Thank you for the comment. We are aware of the limitations of indicator 15.3.1 and the need to find better ways to refine the assessment of land degradation. In particular we have sensitivities on the issues of spatial resolution and the rules that underlie the calculation of the indicator (e.g. more weight should be given to signs of recent degradation and specific thresholds and rules should be used for different spatial contexts..), however the work presented is not intended to be an assessment of the reliability of indicator 15.3.1 but rather a presentation of a tool (S-DSS) that coupled with degradation indicators can be very useful in land management and planning. In using the indicator we realized some specific limitations and tried to mitigate them.  However, we have considered the reviewer's suggestion and added/modified the text as follows:

L505-513 we added the sentence: “This case study gives us the opportunity to stress how important it would be especially at local scale, to calculate and eventually validate the indicator using as base data high resolution and/or additional information (e.g. local dataset, LUCAS dataset). At the same time it would be crucial to both include with greater weight in the calculation, recent changes in land cover (e.g. recent urbanization) and also consider local set of productivity thresholds to address individual land threats as suggested by [20]. In such cases (e.g. peri-urban areas with urbanization intermixed with green areas) new data and rules could significantly change the results of the computations by highlighting local issues that currently are overlooked.”

L533 – 541 original: Nevertheless, major limitations must be considered, including the difficulty of obtaining an up-to-date database on Land Cover and SOC data and the excessive sensitivity of the model to NDVI changes due to the current thresholds of the Trends.Earth methodology. Indeed, to start with, these problems may lead to an underestimation of the SOC variations which can occur very quickly and with an irregular spatial distribution that makes them only detectable through in situ sampling or very high-resolution multispectral image classification.

L533 – 541 reviewed: Nevertheless, major limitations must be considered, including the difficulty of obtain-ing an up-to-date database on Land Cover and SOC data and the excessive sensitivity of the model to NDVI changes due to the current thresholds of the Trends.Earth methodology. The last stress.the need to consider new rules in the indicator calculation such as add specific set of thresholds for local productivity. Indeed, to start with, these problems may lead to an underestimation of the SOC and productivity variations which can occur very quickly and with an irregular spa-tial distribution that makes them only detectable through in situ sampling or very high-resolution multispectral image classification.

  1. L506. How does it set the basis?

We corrected to be more careful as follows:

Original: “this work sets the basis for methodological improvement and comparison.”

Reviewed: “this work sets the basis for some methodological improvements and comparison.”

  1. L515-520. Please rephrase, its not clear.

We checked and rephrased as follows:

Original: Second, the sensitivity of the NDVI model leads to an overestimation of the areas that show improvement (especially in those areas that exhibit, in the reference period, an increase in their vegetation coverage that is generally imperceptible from aerial images, but which modifies their spectral behavior due to climatic factors or soil moisture issues) or degradation (in areas that show decreasing, but temporary, vegetation coverage due to temporary periods of drought).

Reviewed: Second, the sensitivity of the NDVI model leads to an overestimation of the areas that show improvement or degradation (especially in denuded areas), due to temporary changes in vegetation coverage linked to temporary periods of abundant rainfall or drought.

  • 22.   L525-526. “it is probable that some small urban green areas affected the signal”. Could some small urban green areas have been created as well?

Following several verifications through true colour satellite images, we think that in most cases small urban green areas such as flower beds or small gardens may cause misclassification of pixels covering much larger areas (in the case of MODIS data 62,500 square meters)

Reviewer 3 Report

Comments and Suggestions for Authors

The paper under consideration addresses the pressing issue of land degradation and introduces a novel Spatial Decision Support System (S-DSS) developed on Geospatial Cyberinfrastructure (GCI).

The abstract and introduction provide a comprehensive overview of the current challenges in addressing land degradation and introduces a novel solution, a Spatial Decision Support System (S-DSS) developed on Geospatial Cyberinfrastructure (GCI). They outline the significance of Land Degradation Neutrality (LDN) as a political agenda and emphasizes the lack of operational tools for effective policy implementation.

 The Materials and Methods section of the paper outlines a detailed and systematic approach for developing the Land Degradation Neutrality (LDN) tool, with a primary focus on the SDG 15.3.1 indicator. one notable gap is the lack of clarity on why the authors specifically chose PostgreSQL and PostGIS for data management, as opposed to other available open-access options. The choice of a database management system is a critical decision that significantly impacts the overall functionality and performance of the system, and it is essential for the authors to provide a clear rationale for this selection.

it's crucial to emphasize the importance of user and stakeholder feedback for evaluating and improving the proposed Spatial Decision Support System (SDSS). The review should underscore that applying the SDSS to a study case, while informative, may not be sufficient to validate the broader applicability and effectiveness of the system..

The paper acknowledges technical limitations in the LDN calculation methodology, specifically citing concerns about the reliability of using NDVI as a proxy indicator for net primary productivity. The authors should clearly discuss these limitations, related to resolution and threshold definitions in the paper . Also their potential impact on the tool's effectiveness emphasized.

 The authors should stress the need for more extensive user feedback to validate its usability across various spatial extents and diverse scenarios.

Comments on the Quality of English Language

The paper is well-written with clear and concise language, demonstrating a good standard of English proficiency.

Author Response

Dear reviewer, thank you for your comments and insights. We will try to consider your suggestions in their whole. Below our comments to your review point by point:

The paper under consideration addresses the pressing issue of land degradation and introduces a novel Spatial Decision Support System (S-DSS) developed on Geospatial Cyberinfrastructure (GCI).

The abstract and introduction provide a comprehensive overview of the current challenges in addressing land degradation and introduces a novel solution, a Spatial Decision Support System (S-DSS) developed on Geospatial Cyberinfrastructure (GCI). They outline the significance of Land Degradation Neutrality (LDN) as a political agenda and emphasizes the lack of operational tools for effective policy implementation.

  • The Materials and Methods section of the paper outlines a detailed and systematic approach for developing the Land Degradation Neutrality (LDN) tool, with a primary focus on the SDG 15.3.1 indicator. One notable gap is the lack of clarity on why the authors specifically chose PostgreSQL and PostGIS for data management, as opposed to other available open-access options. The choice of a database management system is a critical decision that significantly impacts the overall functionality and performance of the system, and it is essential for the authors to provide a clear rationale for this selection.

In this respect we have included a new reference [22] to a very recent publication dedicated to the LANDSUPPORT platform and the logic behind the development of the SDSS, its framework including database (and management), server and graphical interfaces,  and all the integrated tools. The manuscript follows the general structure of previous works dedicated to other SDSS tools published in various journals. Moreover, our manuscript focuses on a system created in 2017 when all the technological choices were made. Specifically, the choose of PostgreSQL + PostGIS was made for the internal management of vector geospatial data, considering that it is a relational DB, it is open-source and it fully implements the Simple Feature standard. The final publication using the OWS standard is performed through GeoServer which is connected to PostgreSQL data sources.

  • It's crucial to emphasize the importance of user and stakeholder feedback for evaluating and improving the proposed Spatial Decision Support System (SDSS). The review should underscore that applying the SDSS to a study case, while informative, may not be sufficient to validate the broader applicability and effectiveness of the system.

Thank you for this comment which gives us an opportunity to clarify some important issues. Our paper describes one among the 100 tools that represent the general S-DSS system [reference 22]. These include tools based on the application of  physically based models (previously calibrated and validated) and tools that rely on empirical approaches. In this regard, the issue of validation and stakeholder involvement is well known and addressed in the previously cited publication [22] where we stress the concept at various steps including:

  • " Our solution is the S-DSS LANDSUPPORT platform, consisting of a free web-based smart Geospatial CyberInfrastructure containing 15 macro-tools (and more than 100 elementary tools), co-designed with different types of stakeholders and their different needs, dealing with sustainability in agriculture, forestry and spatial planning”

  • “In our case, co-development included different forms of feedback to the developers, depending on the issues raised, such as (i) semi structured interviews, including requests and remarks from experts; (ii) e-mails describing problems in using tools and (iii) direct interaction with the developers. The feedback activities contributed to the co-development and co-creation of the S-DSS tools by identifying the main concerns of stakeholders concerning the tools”.

In the present work we are aware of the need to validate and improve the data and the output. For example, between L90-95 and L496-498 we report the following:

  • " The occurrence of many detailed studies which report limitation in LDN estimates demonstrates the increasing importance of land degradation in science. More specifically, over the last two decades, there has been an exponential increase in scientific publications on land degradation assessment methodologies and new approaches [20]. Thus, the entire research area is very dynamic and new SDG15.3 approaches must be able to adopt any new improvement quickly once it has been properly validated."

  • “The system is flexible and open to further implementation of both models and databases”.

All the above allows us to stress a concept. We did not produce new data with the LDN tool but used an official indicator applied throughout EU, the subject of numerous publications. Attentive to the principle of validation, we improved the indicator with objective data by masking densely populated urban areas, after which we matched it with other EU official layers such as Corine Land Cover data to provide not solution but information. In conclusion, the LDN tool operation is unique but not very complex, then it required easy technical/process validation. On the other hand, the indicator 15.3.1 certainly requires greater attention and stakeholders involvement in-depth analysis on validation and calibration, but these topics are not the specific focus of this paper.  

  • The paper acknowledges technical limitations in the LDN calculation methodology, specifically citing concerns about the reliability of using NDVI as a proxy indicator for net primary productivity. The authors should clearly discuss these limitations, related to resolution and threshold definitions in the paper . Also their potential impact on the tool's effectiveness emphasized.

Recalling the concepts expressed in the previous comment, the objective of this paper was not to analyse in detail the reliability of the indicator 15.3.1 but rather to present a tool integrated into an S-DSS that would use the official indicator to provide thematic information rather than solutions. During the operational analysis we became aware of important issues for the specific class of urbanized areas and for these and only these we made an improvement in the data by modifying the output of the indicator. In this regard, we found two major limitations: i) the thresholds applied by the official procedure (Trends.Earth) in classifying with the sub-indicator "productivity" the areas in terms of degradation/non-degradation, are excessively narrow which results in an excessive sensitivity to changes in NDVI trends over time; ii) the spatial resolution of MODIS data is coarse in urban environments where small, scattered green areas can affect NDVI values leading to overclassification of areas under improvement; These concepts are clearly expressed between L257-269 (L269-284 reviewed version) where we added the sentence: "These evidences are somehow confirmed in a work published on an evaluation of the indicator 15.3.1 [20] in which the authors propose conduction of case studies with high resolution data and setting of local thresholds..”

Moreover, discussing the case study 3 centred on urban environment, we added (L508-516 reviewed version) the sentence: “This case study gives us the opportunity to stress how important it would be especially at local scale, to calculate and eventually validate the indicator using as base data high resolution and/or additional information (e.g. local dataset, LUCAS dataset). At the same time it would be crucial to both include with greater weight in the calculation, recent changes in land cover (e.g. recent urbanization) and also consider local set of productivity thresholds to address individual land threats as suggested by [20]. In such cases (e.g. urban and peri-urban areas with urbanization intermixed with green areas) new data and rules could significantly change the results of the computations by highlighting local issues that currently are overlooked.”

  • The authors should stress the need for more extensive user feedback to validate its usability across various spatial extents and diverse scenarios.

Please, consider our reply to your question number 2 which quotes “It's crucial to emphasize the importance of user and stakeholder feedback…”

Round 2

Reviewer 2 Report

Comments and Suggestions for Authors

Comments on the Quality of English Language

Author Response

Dear reviewer, thank you for your insights. We will try to improve again our work following your suggestions. Please, below our responses to your comments (ref. file: land-2733143_RevPM2):

  • The abstract is indeed improved in the second version, but please consider rephrasing a bit again in order to make it clear that the S-DSS (www.land-support.eu) is an already developed and running system and the purpose of this work is to present the a new tool, the LDN.

Abstract L19 -24: we rephrased “It is within this framework that this paper attempts to demonstrate a new  valuable web-based operational LDN tool as component of an already running Spatial Decision Support System (S-DSS) that is developed on a Geospatial Cyberinfrastructure (GCI). The tool could be offered to EU administrative units (e.g. municipalities) so that they may better evaluate the state and the impact of land degradation in their territories.”

L150: we added “already running”

  • In regards to general comment 2 from round one, i.e., scalability issues over Europe and the provided answers from the authors, it is quite important to include some clear remarks on the discussion elaborating on this, including issues related to the application of the indicator and its components on different scales but also the definition of degradation and improvement over large extents, potentially also including the issue raised on comment from L196-198, L220-222. and bush encroachment in south Europe.

  • In regards to Round 1 comments 15-16 and respective answers for the urban, peri-urban and urban green, please consider including a/some respective phrase/s on the Discussion section as well.

Thank you to address this issue in your two comments above, however, the work presented is not aimed to analyse in detail weaknesses and strengths of the indicator, but rather to take advantage of what is currently available to improve its usability. However, we have tried to cite the issue by rewording the part between L612 and L628 in the discussions as follows: “The LDN tool effectively provides the knowledge base for end-users to understand the soil degradation situation in specific territories and allows a multilevel comparison between different territories, in order to improve awareness among stakeholders and   ordinary citizens who want reliable and comparable data on the subject. However, it is necessary to point out that in its current form the SDG 15.3.1 indicator has greater applicability at the territorial scale than at the local scale. This emerges clearly for example from our analyses in urbanized environments, where we realized that green areas intermixed with sealed surfaces may not be "seen" by the indicator when sparse and scattered, or may cause misclassifications due to edge effect when large and more dense. This is clearly an issue of spatial resolution of the indicator components. Not surprisingly, results for the SDG 15.3.1 indicator describe a land degradation situation in Europe that needs to be investigated in finer thematic detail and geometric resolution. One solution to these issues could be to use different data, with different spatial resolution, to calculate the indicator at different scales. For example, application at the local scale could include the use of Landsat or Sentinel data for the calculation of productivity metrics”

  • In relation to above comments 2-3 you may also consider forming your Conclusion section to describe clearly what this paper suggests and achieves and move more general comments and discussing parts to the discussion section.

Thank you for suggesting to improve again, however, we believe that considering the real objective of the paper, i.e. describing the functionality of a tool integrated into a complex SDSS structure made up of over 100 tools, the conclusions, although synthetic, are exhaustive and well-integrated. In fact, the limits of the indicator are cited both in terms of calculation methodology and spatial resolution of the data. It is underlined how the modular SDSS system can be integrated with new data if available (i.e. a new version of the indicator), the limits of the tool are cited too, such as the achievements of the paper.

  • Upon discussion with the editor you could consider including the additional materials (appendix A) on the main manuscript on Results.

Thanks for the suggestion and interaction with the editor. We have considered this opportunity by including the contents of Appendix A among the results section. This involved adapting the text (L336-386) and renumbering the figures;

 Some detailed comments below:

  • L121-128: Please also check if it is relevant for you to include some of this works/tools as well: https://www.sciencedirect.com/science/article/pii/S0303243419311869?via%3Dihub https://www.sciencedirect.com/science/article/pii/S1462901119302308?via%3Dihub https://geoessential.unepgrid.ch/mapstore/#/dashboard/4 https://projectgeffao.users.earthengine.app/view/reu-ldn-assessment https://wocatapps.users.earthengine.app/view/ldn-prais4

And it would be for sure better to discuss and describe the available webgis tools/works on a previous paragraph before the one starting on L117, that denoted your work, and then rephrase L121 to something like “In fact currently there are WEBGIS aiming to visualize land degradation maps, as described above, but there is not a SDSS 128 tool enabling a customized LDN reporting at any NUTS level for the entire EU territory”

Thanks for the suggestions. We have reshaped the text by making important changes between L121-168 as follows:

“In general terms, it can be stated that there is awe need to improve SDG 15.3.1 assessment approaches so enabling us to achieve both feasible, easy, sustainable uploads of new data/model produced by better scientific approaches or simply by a better local dataset (UNCCD) and easy communication towards large public and public bodies through easy SDG 15.3.1 data ac-cess, visualization and reporting at any local, regional, national level.

The good news is that the research community is moving in these directions, as evidenced by numerous and recent scientific papers devoted to the issue, such as [21] by which the authors propose a proof of concept for a scalable and flexible approach to monitor land degradation at various scales (e.g., national, regional, global) using various components of the Global Earth Observation System of Systems (GEOSS), or [22] in which the authors address the reliability of UNCCD indicators results when compared with percep-tion-based studies, and the implications this has for setting and monitoring land degradation.

Interesting FAO projects also addresses the issues of scalability and resolution through the creation of web-gis systems in which the main goal is to allow easy visualizations of global SDG products based on different algorithms, raising awareness about the need to create products at national level that fit local conditions (https://projectgeffao.users.earthengine.app/view/reu-ldn-assessment; https://wocatapps.users.earthengine.app/view/ldn-prais4).

Other examples of such web-gis systems are LandGis (https://opengeohub.org/article/pre-release-landgis/) presented as the “Open-StreetMap for land-related environmental data”, Queensland soil and land re-source data web map service (https://www.data.qld.gov.au/dataset/queensland-soil-and-land-resource-data-web-map-service) which contains soil site data and soil polygon mapping data which is represented as four main types: soil, land management and land degradation mapping, Land Portal (https://landportal.org/book/indicator/un-aglnddgrd) expressly dedicated to several land indicators including the SDG 15.3.1,

However, Iin this entire domain, there is a clear knowledge gap between the potential of the SDG15.3.1  indicators and their actual use in land planning and management. In addition, the potential use of these indicators is limited if they are not integrated into operational support systems which produce spatially explicit information and provide more structured indica-tions (e.g. location and causes of and use of the land affected by the potential degradation) to better implement environmental policies.

Based on this background, the work proposed has a double objective: (i) a re-liability assessment of the Trends.Earth approach limited to urbanized areas, followed by a potential improvement in the SDG 15.3.1 indicator and (ii) the development of an operational LDN tool to enhance SDG 15.3.1 applicability towards an extensive range of potential users. In fact currently there are WEBGIS aiming to visualize land degradation maps, as described above,  but there is not  a SDSS tool enabling a customized LDN reporting at any NUTS level for the entire EU territory.”

  • L189-192. Still there should be a phrase denoting that the anomalies were solved, not giving much details, but rather stating the main issues and general approach in solving, both for raster and vector data.

We verified that the reported sentence referred more to the general SDSS LANDSUPPORT than to the LDN tool. Therefore, we rephrased the sentence as follows.

“Before being integrated into the database, all  spatial data were checked for potential anomalies. Vector layers (polygons) were firstly checked and eventually corrected in GIS environment for geo-referencing and data  anomalies (i.e. spatial coordinates, missing data, outliers) and then loaded into the geospatial database, which allows location queries (run in SQL)”.

  • L241-242: The authors reply denoting that you they have dealt with this issue in chapter 3.1., how is related to detected anomalies? It seems it’s related only to the reliability issue.

This comment is related to the previous. Since the raster anomalies in the case of the LDN tool were basically related only to geo-referencing and/or projections, we specified adding “(i.e. geo-referencing, projections) L292.

  • L219-220: The use of ‘Homogeneous’ here maybe misleading to be referring to the LC classes of the area, whereas the authors maybe mean the ‘same product’ coverage in terms of area extent and time span ? Please explain or rephrase
  • L220-L222: Maybe it would be more clear to rephrase to something like “Corine Land Cover maps for 2000 and 2018 220 (CLC), in raster format with 100 m cell size, were used, using as land cover legend/classes for the indicator the CLC first level classes of Forest and semi-natural areas, Agricultural areas…

Here we referred to LC classes. We reshaped the sentence as follows: “To this end were used Corine Land Cover maps for 2000 and 2018 (CLC), in raster format with 100 m cell size, using as common land cover legend/classes for the indicator the CLC first level classes of Forest and semi-natural areas, Agricultural areas, Wetland, Artificial surfaces and Water bodies

Some Syntax – Typos below:

  • L22 maybe 'On-The-Fly Computing' ?

Accepted

  • L33 ‘table’ tables ?

Accepted

  • L174,326 etc consider changing from ‘user’ to ‘users’ and from ‘his’ to ‘their’

Accepted

  • L194-195 bad syntax

We reformulated the sentence as follows: The most important layer for the LDN tool is the land degradation map, which was chosen  to assess the state of land for the entire European territory through the SDG 15.3.1 indicator”

  • L258 bad syntax

We reformulated the sentence as follows: “Google Earth satellite orthophotos observed with the time scrolling tool were used as further verification that there were no changes in the areas chosen during the period covered by the indicator.

  • L520 bad syntax (e.g., ‘has been highlighted’ at the end of the sentence, and omit ‘it’)

Accepted

  • L525 bad syntax (e.g., ‘has been highlighted’ at the end of the sentence, and omit ‘it’)

Accepted

  • L567 bad syntax (e.g., ‘has been described’ at the end of the sentence, and omit ‘it’)

Accepted